# Validation and Analysis of the Polair3D v1.11 Chemical Transport Model Over Quebec

Shoma Yamanouchi[1], Shayamilla Mahagammulla Gamage[1], Sara Torbatian[1], Jad Zalzal[1], Laura Minet[2], Audrey Smargiassi[3], Ying Liu[3], Ling Liu[4], Forood Azargoshasbi[2], Jinwoong Kim[5], Youngseob Kim[6], Daniel Yazgi[7], Andrée-Anne Brown[8], and Marianne Hatzopoulou[1]

[1]University of Toronto, Department of Civil and Mineral Engineering, Toronto, Ontario, Canada
[2]University of Victoria, Victoria, British Columbia, Canada
[3]Université de Montréal, Montreal, Quebec, Canada
[4]Environmental Health Science and Research Bureau, Health Canada, Ottawa, Ontario, Canada
[5]Meteorological Research Division, Environment and Climate Change Canada, Dorval, Quebec, Canada
[6]CEREA, École des Ponts, EDF R&D, Marne-la-Vallée, France
[7]Swedish Meteorological and Hydrological Institute, Folkborgsvägen, Norrköping, Sweden
[8]Ministère de l'Environnement et de la Lutte contre les changements climatiques, Quebec City, Quebec, Canada

**Correspondence:** Shoma Yamanouchi (shoma.yamanouchi@mail.utoronto.ca)

**Abstract.** Air pollution is a major health hazard, and while air quality overall has been improving in industrialized nations, pollution is still a major economic and public health issue, with some species, such as ozone ($O_3$), still exceeding the standards set by governing agencies. Chemical transport models (CTM) are valuable tools that aid in our understanding of the risks of air pollution both at local and regional scales. In this study, the Polair3D v1.11 CTM of the Polyphemus air quality modeling platform was set up over Quebec, Canada to assess the model's capability in predicting key air pollutant species over the region, at seasonal temporal scales and at regional spatial scales. The simulation by the model included 3 nested domains, at horizontal resolutions of 9km by 9km, 3km by 3km, and two 1km by 1km domains covering the cities of Montreal and Quebec. We find that the model captures the spatial variability and seasonal effects, and to a lesser extent, the hour-by-hour or day-to-day temporal variability for a fixed location. The model at both the 3km and the 1km resolution struggled to capture high frequency temporal variability, and showed large variabilities in correlation and bias from site to site. When comparing the biases and correlation at a site-wide scale, the 3km domain showed slightly higher correlation for carbon monoxide (CO), nitrogen dioxide ($NO_2$) and nitric oxide (NO), while ozone ($O_3$), sulfur dioxide ($SO_2$) and $PM_{2.5}$ showed slight increases in correlation at the 1km domain. The performance of the Polair3D model was in line with other models over Canada, and comparable to Polair3D's performance over Europe.

15  *Copyright statement.*

# 1 Introduction

Air pollution is a major health hazard that affects millions of lives globally, and is seen as one of the largest contributors to global disability-adjusted life-years (GBD 2015 Risk Factors Collaborators, 2016). While air quality overall has been improving in Canada, some species, such as ozone ($O_3$), still regularly exceed the standards set by governing agencies (e.g., Ministry of the Environment and Climate Change, 2016). Furthermore, the Canadian government (Health Canada, 2022a) estimated in 2019 that the economic impacts of air quality related health risks are over100 billion Canadian Dollars per year, and that air pollution is linked to 15,300 premature deaths every year in Canada.

Industrial and traffic emissions play a large role in determining urban air quality (e.g., Rai, 2016; Batisse et al., 2017; Wallington et al., 2022; Health Canada, 2022b). In the province of Quebec in Eastern Canada, 410 premature deaths were attributed to traffic related air pollution in 2015 (Health Canada, 2022b). Quebec sees higher levels of particulate matter than the national average, and similar results for nitrogen dioxide ($NO_2$) and sulfur dioxide ($SO_2$). Additionally, industrial emissions and proximity to industrial facilities in Quebec have been associated with adverse health outcomes such as asthma onset in childhood (Buteau et al., 2020), short-term risk of hospitalization in children (Brand et al., 2016) and a decrease in lung function (Smargiassi et al., 2014). As opposed to traffic emissions which mainly take place in densely populated areas, high-emitting industries in Canada are also found in rural areas (Jeong et al., 2011). These regions typically do not have other major sources of air pollution, which results in large gradients in pollution levels in nearby communities.

Environment and Climate Change Canada (ECCC) operates about 250 air pollutant monitoring stations as part of their National Air Pollution Surveillance (NAPS) program (NAPS, 2016), of which 131 are in Quebec (and some may only be reporting limited time periods and/or limited pollutant species). Given the size of the country, and the province, this is far too sparse to be useful in conducting spatial variability analyses of air pollutants. Modeling the sources, chemistry, dynamic transport of atmospheric pollutants is crucial in understanding tropospheric pollution events and mitigating health impacts by identifying affected regions and sensitivity to various emissions. There are mainly two Chemical Transport Models (CTM) used in Canada, GEM-MACH run by ECCC, and USEPA's Community Multiscale Air Quality Modeling System (CMAQ). While the ECCC GEM MACH has been used to model the atmosphere over Canada including Quebec (Chen et al., 2020; Health Canada, 2022b), we attempt to assess the performance of and validate the Polair3D CTM of the Polyphemus air quality modeling platform (Mallet et al., 2007) coupled with emissions derived from both ECCC and the United States emission inventories using the Sparse Matrix Operator Kernel Emissions (SMOKE) emissions-processing system, for key pollutant species: carbon monoxide (CO), $O_3$, $NO_2$, NO, $SO_2$ and particulate matter smaller than 2.5 μm in diameter ($PM_{2.5}$). The Polair3D model has seen little use over North America and particularly over Canada, aside from one example over Ontario, Canada (Minet et al., 2021) and a coarse-resolution study covering all of North America by Sartelet et al. (2012). Unlike CMAQ, which is mainly used at larger, regional scales at coarser resolutions (i.e., horizontal resolutions higher than $\sim$1km$^2$), Polair3D is known to be robust at these higher resolutions (Thouron et al., 2017), and we aim to assess the model performance at these higher resolutions as well as at coarser resolutions with larger modeling domains. In this study, we aim to present a novel use of this model over Quebec,

Canada, using a longer modeling period and a larger modeling domain, to assess the ground (surface) level model performance
at seasonal temporal scales and at regional spatial scales.

## 2   Methods

### 2.1   Model Setup

The Polyphemus platform (Mallet et al., 2007) was used for this analysis. Polyphemus is an open source suite of models developed at the Centre d'Enseignement et de Recherche en Environnement Atmosphérique (CEREA), and in this study, Polair3D
(Sartelet et al., 2002; Mallet and Sportisse, 2004; Pourchet et al., 2005; Boutahar et al., 2004), a CTM within the Polyphemus platform, was utilized. The newest version (v1.11) of the model (Kim et al., 2023) was used, with an updated aerosol chemistry module called SSH-aerosol (Sartelet et al., 2020). This module combines SCRAM (Size-Composition Resolved Aerosol Model), which simulates the dynamics and the mixing state of atmospheric particles, SOAP (Secondary Organic Aerosol Processor), which models the partitioning of organic compounds, and $H^2O$ (Hydrophilic/Hydrophobic Organics), which simulates
the formation of semi-volatile organic compounds formed via the oxidation of Volatile Organic Compounds (VOCs). Polair3D is a Eulerian atmospheric CTM, and includes preprocessing modules for formatting and creating binary input files for meteorology, biogenic emissions, surface deposition, and initial/boundary conditions. Anthropogenic emissions inventories will be discussed later in Section 2.2. For calculating biogenic emissions and deposition at the surface, land-use data from GLC2000 was used (Bartholomé and Belward, 2005).

The meteorology field was taken from pre-run WRF data (NCAR, 2023). The modeling domain comprises four domains that have 27km, 9km, 3km, and 1km grid spacing, respectively, with two-way nesting. The number of vertical levels is 42 spanning from the surface to 100 hPa. Initial and lateral boundary conditions of meteorology were provided by the North American Regional Reanalysis (NARR) (Mesinger et al., 2006) which is available at a 32km grid spacing with 30 vertical levels. Each 30-hour forecast was initialized every 00:00 UTC and had a 6-hour spin-up time. Thus, the first 6 hours of forecasts
are discarded and replaced with forecasts initiated with the previous cycle for overlapping times. Grid nudging was applied for horizontal wind, temperature, and humidity for vertical levels above the planetary boundary layer (PBL) height in the largest domain. Parameterization schemes used in the simulation are as follows: Purdue Lin scheme (CHEN and SUN, 2002) for microphysics, the Rapid Radiative Transfer Model for GCMs (RRTMG) Shortwave and Longwave Schemes (Iacono et al., 2008), Mellor–Yamada–Janjic PBL scheme (Janjić, 1994), Grell–Devenyi ensemble scheme (Grell and Dévényi, 2002) for cumulus
parameterization which was applied only to domain 1 and 2, Unified Noah land surface model (Chen and Dudhia, 2001), and a 3-category urban canopy model (Chen et al., 2011) for urban areas. WRF temperature, as well as wind speed/direction was compared against observational data from meteorological stations operated by ECCC prior to using them in Polair3D.

From this WRF configuration, meteorology from the 9km, 3km, and 1km domains were used. The model configuration is as follows: The model was run in 3 nested domains; the largest and coarsest-resolution domain was roughly 9km by 9km grid-cell
resolution (edges), and within it, a smaller domain of about 3km by 3km resolution was run, and lastly, 1km by 1km resolution runs were performed over the cities of Montreal and Quebec. The modeling domains are shown in Figure 1 (note that parts of

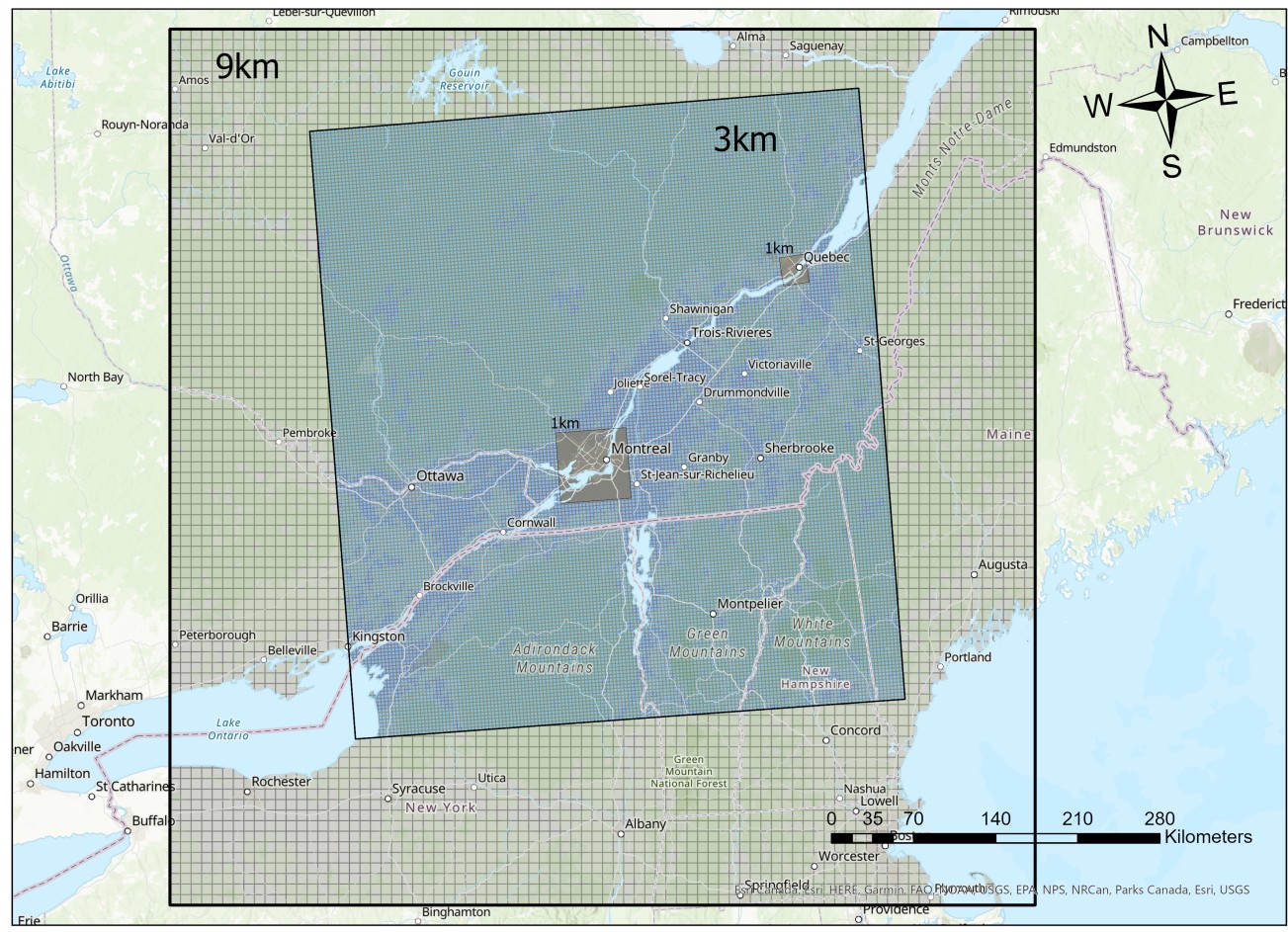

**Figure 1.** The modeling domains used in this study.

the 9km and 3km domains include the United States (US)). The model was run for four seasons out of 2018, with four weeks per season (January for winter, April for spring, July for summer, and October for fall), for a total of 16 weeks of model data. Spin-up was done for 1 week for each run. Boundary conditions for the outermost domain, and the initial conditions for each of the runs were derived from CAM-Chem assimilated data (Tilmes et al., 2015).

The model was run with a 10-minute time step, and output was averaged and saved hourly. The model was run with vertical grids going up to 6000m, but only data from the lower-most layer (surface) was saved. The vertical resolution is as follows: 0m, 20m, 40m, 90m, 150m, 250m, 400m, 800m, 1500m, 2400m, 3500m, and 6000m. For 1km resolution runs, the small domain size led to numerical instabilities, and the time steps were lowered to 5 minutes and 1 minute for Montreal and Quebec City, respectively. In this study, CO, $O_3$, $NO_2$, NO, $SO_2$ and $PM_{2.5}$ from the model were examined, although data for other species were also saved.

## 2.2 Emissions

A Sparse Matrix Operator Kernel Emissions (SMOKE) emissions-processing system was used to prepare the Polair3D emissions input files (CMAS-SMOKE). Emission processing involves three major steps: spatial allocation, temporal allocation, and chemical speciation. Canadian and US emissions in the domain were calculated based on SMOKE-ready formats of the Canadian emission inventory (Sassi et al., 2021) and US National Emission Inventory (EPA: Emissions Modeling Platforms), along with their temporal allocation and chemical speciation data. Spatial allocations for the three nested domains were generated using both Canadian and US spatial allocator inputs (CMAS-SA; CMAS-DB).

Canada's Air Pollutant Emissions Inventory (APEI) also known as the Canadian criteria-air-contaminants (CAC) emissions inventory, is prepared and published by ECCC. The APEI is a comprehensive inventory of anthropogenic emissions of 17 air pollutants including CO, ammonia ($NH_3$), nitrogen oxides ($NO_x$), $PM_{2.5}$, particulate matter smaller than 10 $\mu m$ in diameter ($PM_{10}$), $SO_2$, and VOCs at the national, provincial, and territorial levels. It is compiled from many different data sources. The APEI is developed by the Pollutant Inventories and Reporting Division (PIRD) of ECCC. The inventory databases compiled by PIRD are modified by the Air Quality Modelling Applications Section (AQMAS) of ECCC for emissions processing with SMOKE. For further details of the SMOKE-ready format of the Canadian 2015 APEI inventory refer to Annex 2 of the 1990-2015 Air pollutant emission inventory report: Environment and Canada (2017).

The US National Emission Inventory (NEI) is the second inventory used in this study. NEI includes emissions for the six criteria air pollutants (CAPs) and 187 hazardous air pollutants. The CAP-related emissions are $NH_3$, CO, Pb, $NO_x$, particulate matter (PM2.5, $PM_{10}$, organic carbon, and black carbon), $SO_2$, and VOC. Data on US emissions are derived in several ways: continuous measurements, estimates based on infrequent source samples, and estimates based on average emission rates. In this study, we use a combination of SMOKE-ready formats of NEI 2014 and 2017 inventories from EPA (EPA: Emissions Modeling Platforms).

The emission sectors nonpoint, on road, and nonroad in both Canadian and US inventories were processed in SMOKE as area sources. The point source sectors in both inventories were processed as either 2-dimension (2D) or 3-dimension (3D) (layered) elevated point source emissions. Emissions from point sources such as airports and mines are considered 2D elevated sources. Industrial emissions (e.g. electric power generation, commercial facilities) are calculated as 3D layered emissions. SMOKE analyzes the stack parameters of each facility as well as the meteorology to determine the layers' emissions. SMOKE accesses the stack parameters such as the height, the diameter, and the temperature directly from the industrial emissions reported in the Canadian and US inventories. Meteorology-Chemistry Interface Processor (MCIP) was used to generate the necessary meteorology files (e.g. `GRID_CRO_2D`, `MET_CRO_2D`, `MET_DOT_3D`) for SMOKE volume emission processing (EPA-CMAQ). There are known aberrations in the ECCC offroad emissions in January. To correct for this, the offroad emissions in January were replaced by those of April (offroad emissions do not vary significantly by season). These aberrations were not seen in any of the other sectors nor in any of the other months.

The Polair3D model contains a Size-Composition Resolved Aerosol Model (SCRAM). Thus, the PM AE6 speciated SMOKE output must be incorporated into an input for SCRAM. This conversion is shown in Figure 2. The size distribution of the PM

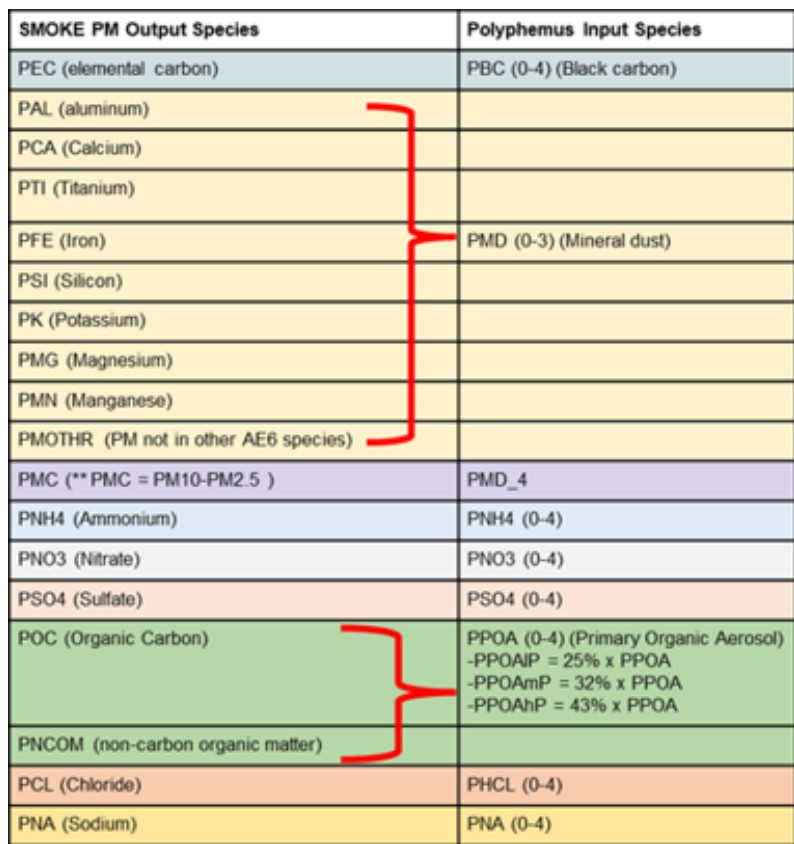

| SMOKE PM Output Species | Polyphemus Input Species |
|---|---|
| PEC (elemental carbon) | PBC (0-4) (Black carbon) |
| PAL (aluminum) | |
| PCA (Calcium) | |
| PTI (Titanium) | |
| PFE (Iron) | PMD (0-3) (Mineral dust) |
| PSI (Silicon) | |
| PK (Potassium) | |
| PMG (Magnesium) | |
| PMN (Manganese) | |
| PMOTHR (PM not in other AE6 species) | |
| PMC (** PMC = PM10-PM2.5 ) | PMD_4 |
| PNH4 (Ammonium) | PNH4 (0-4) |
| PNO3 (Nitrate) | PNO3 (0-4) |
| PSO4 (Sulfate) | PSO4 (0-4) |
| POC (Organic Carbon) | PPOA (0-4) (Primary Organic Aerosol) -PPOAlP = 25% x PPOA -PPOAmP = 32% x PPOA -PPOAhP = 43% x PPOA |
| PNCOM (non-carbon organic matter) | |
| PCL (Chloride) | PHCL (0-4) |
| PNA (Sodium) | PNA (0-4) |

**Figure 2.** SMOKE PM output conversion to Polair3D PM input

species was applied based on the SNAP (Selected Nomenclature for Air Pollution) sectors. The SNAP sectors include combustion in energy and transformation industries, non-industrial combustion plant, combustion in the manufacturing industry, production processes, extraction and distribution of fossil fuels and geothermal energy, solvent and other products, road transport other mobile sources, and machinery, waste treatment and disposal, and agriculture (EMEP/EEA, 2019). A 5-bin size
distribution was applied to each species that was derived from the 10-bin SNAP size distribution values consistent with the Polair3D inputs. For the Polair3D model, primary organic aerosol (POA) species also need to be divided into three categories based on volatility. Hence, first, the POA species were divided into low (POAlP), medium (POAmP), and high volatility (POAhP) and then we applied the 5 bin size distribution for each subspecies.

### 2.3 Surface Observations

Model results were compared against surface observations to validate the model and assess its performance in modeling key air pollutant species at the surface level. Surface observations collected as part of the National Air Pollution Surveillance (NAPS) program (NAPS, 2016) were used in this analysis.

The air pollution species examined in this study (CO, $O_3$, $NO_2$, NO, $SO_2$ and $PM_{2.5}$) are reported by some, but not all NAPS sites; many NAPS sites only report some of the species, and some sites may not have data during the modeling time period.

To assess the modeling performance, several statistics were examined: Pearson correlation coefficient ($R$), mean relative difference (MRD), mean squared error (MSE), mean bias (MB) and normalized mean bias (NMB). MRD, MSE, NB and NMB were calculated by subtracting NAPS from the model (i.e., $\text{MRD} = 100. \times \frac{\text{Model}-\text{NAPS}}{\text{Model}}$, $\text{MSE} = \mathbb{E}[(\text{Model} - \text{NAPS})^2]$, $\text{NB} = \frac{\Sigma[\text{Model}-\text{NAPS}]}{N}$ and $\text{NMB} = 100. \times \frac{\Sigma[\text{Model}-\text{NAPS}]}{\Sigma[\text{NAPS}]}$ ). These statistics were chosen following Emery et al. (2017).

    Additionally, for $NO_2$ and $PM_{2.5}$, the model was compared against assimilated monthly ground level National LUR (land use

regression) dataset products from Canadian Urban Environmental Health Research Consortium (CANUE) (for $NO_2$) (Hystad et al., 2011; Weichenthal et al., 2017; DMTI Spatial Inc., 2015) and Atmospheric Composition Analysis Group (ACAG) for ($PM_{2.5}$) (van Donkelaar et al., 2021). The CANUE $NO_2$ dataset was developed from 2006 NAPS data using land use regression model taking into account various geographic variables (road length within 10 km, area of industrial land use within 2 km and summer rainfall) and satellite data (from 2005 to 2011) (Hystad et al., 2011). In this dataset, monthly averages of $NO_2$ are given

for each Canadian postal code (DMTI Spatial Inc., 2015), and the comparison analysis with the model was done using the 3km resolution model by binned into the closest model grid cell. 2018 monthly dataset was used. ACAG $PM_{2.5}$ dataset is derived by assimilating aerosol optical depth (AOD) retrievals from the NASA MODIS, MISR, SeaWIFS, and VIIRS instruments with the GEOS-Chem chemical transport model, and was calibrated to global ground-based observations using a geographically weighted regression (van Donkelaar et al., 2021). The dataset has a resolution of $0.01°$ by $0.01°$. The comparison analysis was

done using the 3km model resolution, and the ACAG dataset was binned into the Polair3D similar to the analysis done with the CANUE National LUR dataset. 2018 monthly dataset was used.

## 2.4   Test Scenarios

As discussed in Section 1, industrial emissions have been associated with adverse health outcomes. Understanding the behavior of the model under various emissions scenarios is important for performing pollution exposure analyses. To enrich the model

validation findings, and to qualitatively assess the behavior of the Polair3D model under varying emissions scenarios, a run with no industrial emissions was performed for the same domain and time frames. Other emissions (such as biogenic and traffic emissions) were kept the same. All other variables and input files, including the meteorology and the model configurations, were kept the same as the base case (i.e., with all emissions).

    Additionally, two test scenarios, one with only the emissions from smelter, refinery and foundry industries suppressed, and

another scenario with only the emissions from paper and pulp industry turned off, were run. These scenarios were adapted from a study by Liu et al. (2024). As with the no industry scenario, all other variables and input files were kept the same as the base case.

## 3 Results and Discussions

### 3.1 3km Resolution

The 3km modeled monthly averages (for January, April, July and October) for the entire domain are presented in Figure 3 for CO, $O_3$, $NO_2$, NO, $SO_2$ and $PM_{2.5}$ (a note about these figures is that if the concentrations are above or below the scale, chosen here to exclude the lowest and the highest percentile, the figures will show white; this was done to preserve important spatial details in the mid-range of the data). For $O_3$, 8-hour maximum daily average (MDA8) was examined as recommended by Emery et al. (2017).

A site-wide analysis, that is, comparing monthly averages across all NAPS sites, resulted in higher correlation for CO, $O_3$ and $NO_2$ than NO, $SO_2$ and $PM_{2.5}$. CO exhibited a high correlation coefficient ($R$) of 0.91 across 28 data points, while NO showed the lowest correlation at $R = 0.27$ (see Table 1). Correlation plots for all species at the 3km resolution can be found in Figure 4. Both $O_3$ and NO showed higher correlation in winter than in the summer, going from $R = 0.83$ in January down to 0.63 in July for $O_3$, and 0.27 to 0.03 for NO. While $NO_2$ correlation did go down over the summer, it was not to this extent

($R = 0.69$ in January, down to 0.53 in July). Sartelet et al. (2012) reported, in their study using the Polair3D model covering all of North America (at a coarser-resolution of 0.25° by 0.25°), $O_3$ correlation of 0.604, comparable to our overall correlation of 0.85. Both modeled $O_3$ and $NO_2$ showed some overestimation bias, with MRD=21.4% and 33.5%, respectively (see Figure 5). Indeed, $NO_2$ and $SO_2$ both showed overestimation biases, despite CAC emissions being thought to underestimate emissions of these species (Krzyzanowski, 2009). Minet et al. (2021) saw large overestimations of $O_3$ in their Polair3D validation effort, and

attributed it to MOZART4 boundary conditions; the boundary conditions in this study were derived from CAM-Chem. Also of note is the result that MRD in $O_3$, a photochemically reactive pollutant that typically peaks in summer time in the troposphere, was similar in both summer and winter (18.2% and 23.5% for July and January, respectively).

NO, on the other hand, showed overestimation overall (see Figure 4d), but the calculated MRD was -59.2% (indicating under estimation), likely resulting from a few very high values seen in the NAPS data. This is similar to the GEM-MACH model

over North America, which also overestimated CO, $O_3$, $NO_2$ and NO (Stroud et al., 2020; Makar et al., 2015). In fact, the GEM-MACH model over Toronto, Canada, was shown to overestimate $NO_2$ to a larger extent than $O_3$, much like the results presented here (Stroud et al., 2020).

$PM_{2.5}$ performance showed mixed results; the bias was relatively small with an overall MRD of -3.9%, but correlation varied significantly from $R = 0.82$ in January, to $R = 0.36$ in July (and 0.45 overall). Minet et al. (2021) also noted the poor

correlation for modeled $PM_{2.5}$ in their study that examined Polair3D performance during a particular summer day (in August). Model performing worse in the summer was a common theme seen in all species except for $SO_2$, which had the opposite trend with $R = 0.57$ in January and 0.66 in July. The correlation is comparable to a study by Sartelet et al. (2012) who reported a correlation of 0.504 for $PM_{2.5}$. Modeled $SO_2$ showed overestimation as well, with MRD=66.5%. Another noteworthy point here is that correlations of CO were, in most cases, better than those of other primary (emitted) pollutants like $NO_2$, NO and

$SO_2$; this may be explained by uncertainties in emissions (Kim et al., 2018).

The model performance was more challenging when looking at individual sites. An example of $O_3$ time series and corre-

lation plots from January are shown in Figure 6; this plot shows the NAPS data and the modeled $O_3$ over a NAPS site in Montreal (NAPS ID: 50135) in July (both the raw comparison and comparison using the MDA8 metric are shown). There was considerable variability in correlation from site to site; this site showed relatively good correlation, especially for the raw comparison. The model showed relatively small biases for $O_3$ and captures the overall ranges seen in observational data (NAPS mean values were within the model mean $\pm$ 1 model standard deviation for most sites). MDA8 $O_3$ comparison generally fared better in winter months (January), with worse correlation in other times of the year; the July plot shows worse correlation with the MDA8 than with the raw comparison.

For all species, there was considerable variability in correlation coefficients from site to site, and resampling (e.g., 6 hour average) the dataset did not lead to improved correlation (up to 48 hour averages were tried in this analysis), although as noted above, for $O_3$, comparison using MDA8 did result in better correlation. NAPS data are reported to the nearest integer values, meaning the data is quite coarse, leading to "discretization" artifacts that can clearly be seen in Figure 6a. These results suggest that the model is better at capturing the spatial variability and seasonal effects, rather than hour-by-hour or day-to-day temporal variability for a fixed location. Furthermore, NAPS sites are categorized into several site-types, including regional background (RB), general population exposure (PE), and transportation-influenced (T), and when looking at correlations from this perspective, the highest $O_3$ correlation was seen with the RB sites that indicate that the model is able to capture the overall amount of background $O_3$ that is generated/destroyed. Correlations for $NO_2$ and NO were poor for PE sites (generally in urban areas); suggesting that there are large uncertainties in emissions.

Comparisons with the National LUR monthly $NO_2$ dataset show relatively good agreements for all four months. $R$ ranged from 0.82 in July to 0.86 in October. The correlation plots can be seen in Figure 7. Although the model mean was higher (i.e., the model is overestimating) for all months except for January, MRD ranged from -48% to -22%. This indicates that in places where the model is underestimating, the model underestimates by a large margin; because MRD calculation involves dividing by the model value, if the model values are small (and smaller than the National LUR values), the MRD becomes a large negative number. The collation against ACAG $PM_{2.5}$ showed relatively worse correlations. Highest $R$ was 0.59 in April, and worse correlation was $R = 0.45$ in October. The correlation plots can be seen in Figure 8. MRD ranged from -80% to -10%, and the model mean was lower than the ACAG mean for all months except for October.

## 3.2 1km Resolution

The 1km model was run over the cities of Montreal and Quebec City (see Figure 1 for the domains). As noted in Section 2, the time step of the model was shortened to 5 minutes and 1 minute for Montreal and Quebec City, respectively, down from 10 minutes, to increase model stability at these small domains (see Section 2.1). The modeled monthly averages (for January, April, July and October) for the entire domains are presented in Figures 9 and 10 (for Montreal and Quebec City, respectively) for CO, $O_3$, $NO_2$, NO, $SO_2$ and $PM_{2.5}$.

When comparing the biases and correlation at a site-wide scale, the higher resolution 1km runs did not result in strictly better performance. Indeed, when analyzing the same sites (i.e., restricting the 3km analysis to the NAPS sites seen in the smaller 1km run), the coarser 3km model showed slightly higher correlation for CO, $NO_2$ and $O_3$, and while $SO_2$ and $PM_{2.5}$

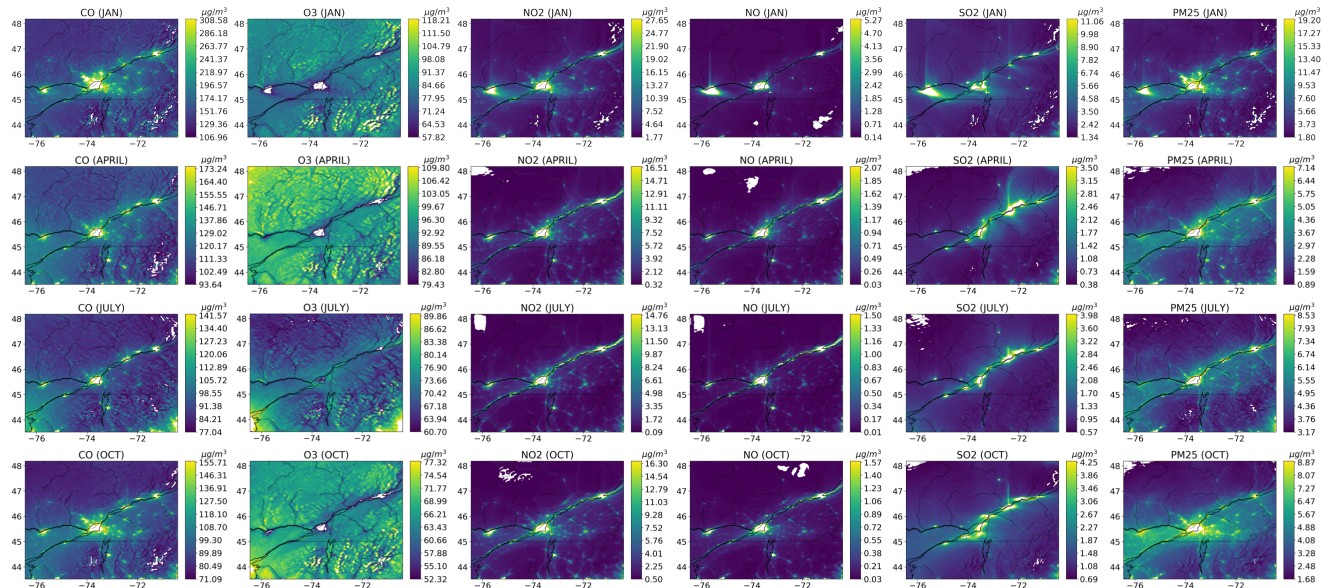

**Figure 3.** Monthly averages (January, April, July and October, from top to bottom) of the Polair3D model at the 3km resolution for (from left to right) CO, $O_3$, $NO_2$, NO, $SO_2$ and $PM_{2.5}$. All units are in µg/m$^3$. Note that if the concentrations are above or below the scale (chosen here to exclude the lowest and the highest percentile), the figures will show white; this was done to preserve important spatial details in the mid-range of the data).

**Table 1.** Site-wide comparison summary table showing correlation (R), mean relative difference (MRD), mean squared error (MSE) (µg/m$^3$), mean bias (MB) (µg/m$^3$), normalized mean bias (NMB), (see Section 2.3 for more detail) and number of data points (N), using data from all four simulation months (January, April, July and October) for the 3km run, 3km run with comparisons restricted to NAPS monitoring sites found in the 1km domain only, and 1km run. For $O_3$, MDA8 was first calculated, and used for this analysis (see Section 3.1 for more detail).

| | 3km | | | | | | 3km with 1km domain sites only | | | | | | 1km | | | | | |
|---|---|---|---|---|---|---|---|---|---|---|---|---|---|---|---|---|---|---|
| Species | R | MRD | MSE | MB | NMB | N | R | MRD | MSE | MB | NMB | N | R | MRD | MSE | MB | NMB | N |
| CO | 0.91 | -12.7 | 1803 | -17.1 | -7.1 | 28 | 0.91 | -13.1 | 1945 | -17.7 | -7.2 | 24 | 0.91 | -13.3 | 2156 | -17.2 | -7.0 | 24 |
| $O_3$ | 0.85 | 21.4 | 466 | 19.5 | 27.3 | 144 | 0.86 | 21.9 | 504 | 20.0 | 28.8 | 72 | 0.70 | 5.69 | 383 | 11.1 | 15.9 | 72 |
| $NO_2$ | 0.70 | 33.5 | 214 | 11.5 | 78 | 75 | 0.67 | 39.3 | 288 | 12.3 | 79 | 67 | 0.41 | 45.8 | 751 | 18.4 | 118 | 67 |
| NO | 0.27 | -59.2 | 30.2 | 1.2 | 22.1 | 72 | 0.25 | -29.1 | 32.4 | 1.2 | 21.5 | 65 | 0.04 | -19.6 | 2844 | 13.9 | 248 | 65 |
| $SO_2$ | 0.52 | 66.5 | 22.7 | 3.4 | 224 | 40 | 0.52 | 66.0 | 27.6 | 3.8 | 209 | 28 | 0.65 | 74.1 | 2427 | 19.4 | 1061 | 28 |
| $PM_{2.5}$ | 0.45 | -3.90 | 22.1 | 1.8 | 26.3 | 123 | 0.39 | 34.0 | 35.5 | 4.7 | 61.7 | 65 | 0.55 | 41.2 | 51.6 | 6.2 | 81.4 | 65 |

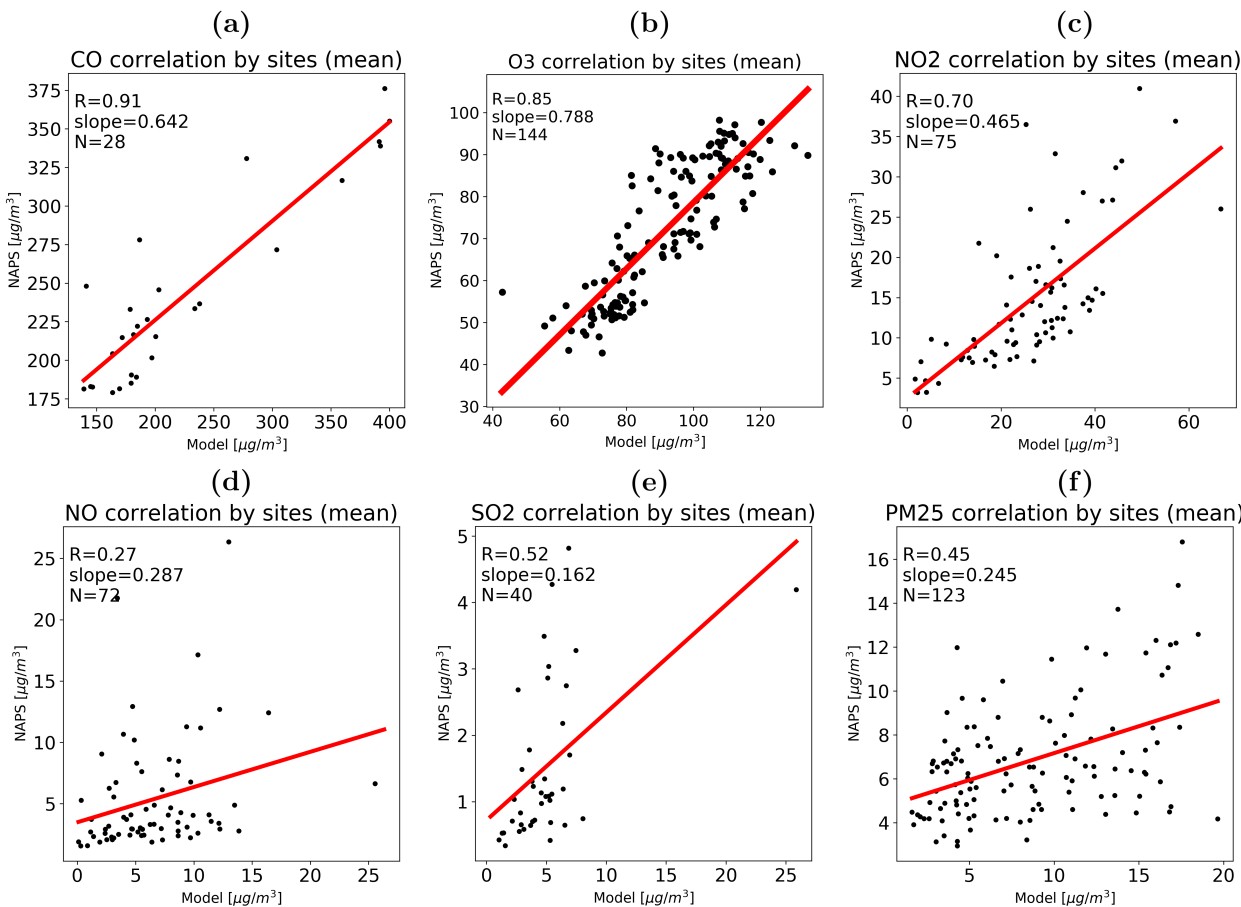

**Figure 4.** Monthly average site-wide correlation plots at the 3km resolution for CO, $O_3$, $NO_2$, NO, $SO_2$ and $PM_{2.5}$, for (a), (b), (c), (d), (e) and (f), respectively. For $O_3$, MDA8 was first calculated, and used for this analysis (see Section 3.1 for more detail). All units are in $\mu g/m^3$.

showed increases in correlation when running at the 1km resolution, the differences were small, for example going from 0.52 to 0.65 (for 3km and 1km, respectively) for $SO_2$ (see Table 1, note that MDA8 metric was used for $O_3$). Examining the model performance site by site showed similar results. Running the model at an increased resolution may be an effective way to downscale the data, but it does not appear to make the simulation more temporally accurate. Similar results were reported by

240    Russell et al. (2019). Their model (GEM-MACH) did not show improvements in "standard scoring methodologies" (such as correlation with surface observation sites) when increasing their model resolution from 2.5km to 1km.

To assess the model performance during the daytime versus nighttime, a similar site-wide analysis was done but this time separating the daytime data and nighttime data. The results can be seen in Figures 11 and 12, for correlation and box plots, respectively. One noteworthy result is that the slope was higher during the day for all species except $SO_2$. Correlation was

245    higher for CO, $O_3$, NO and $PM_{2.5}$, and was slightly lower for $NO_2$ and $SO_2$. Furthermore, $O_3$, a secondary pollutant that is

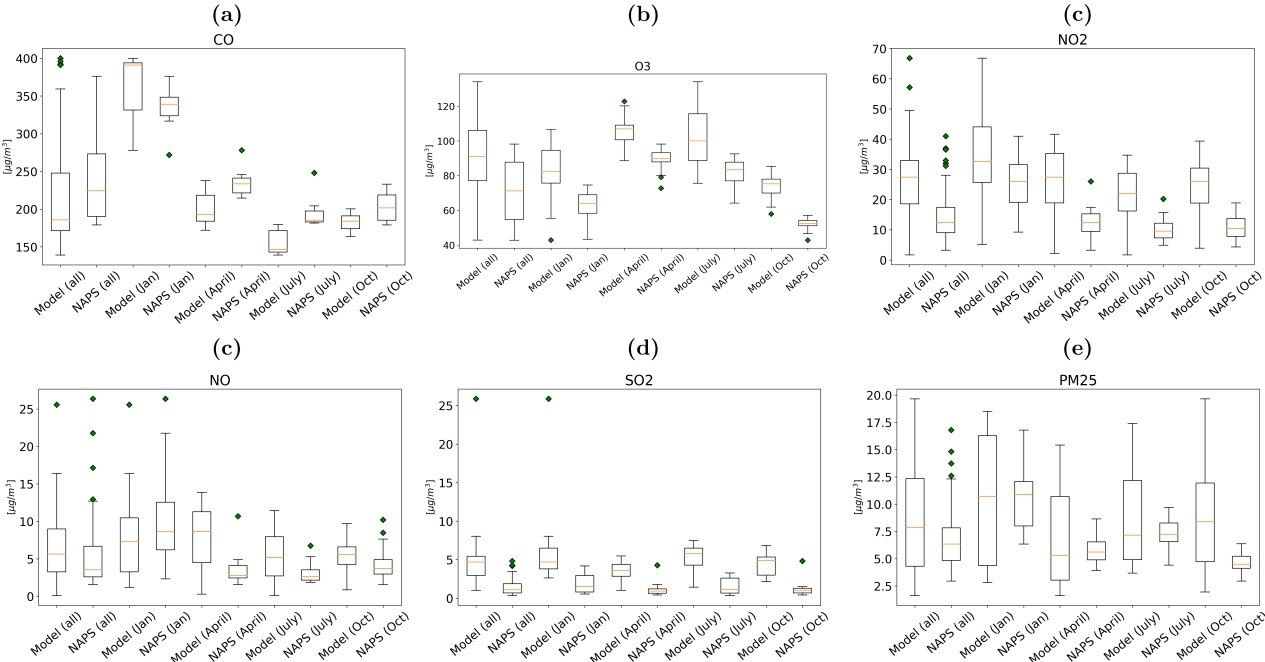

**Figure 5.** Monthly average site-wide comparison box plots at the 3km resolution for (a) CO, (b) $O_3$, (c) $NO_2$, (d) NO, (e) $SO_2$ and (f) $PM_{2.5}$. For $O_3$, MDA8 was first calculated, and used for this analysis (see Section 3.1 for more detail). All units are in $\mu g/m^3$.

created and destroyed photochemically and thus heavily affected by sunlight, showed higher correlation during the day than night (see Figure 11 correlation plot), and at the same time showed large underestimation biases during the night (See Figure 12 boxplot). This suggests that the model is capable of modeling $O_3$ during the day but struggles to simulate the background $O_3$ during the night when photochemical reactions are low and/or nonexistent. For CO, correlation was significantly higher during the day than night ($R = 0.73$ versus -0.69), although the difference was less extreme when looking at individual months.

### 3.3 Comparison with Other Models

Polair3D performance over Canada is in line with models, such as GEM-MACH, over Canada. A study by Russell et al. (2019) which examined the performance of GEM-MACH over Alberta, Canada at both 1km and 2.5km resolutions, saw similar correlations. In their study, they calculated correlation coefficients for $O_3$, $SO_2$, and $PM_{2.5}$ to be 0.496 (0.506 for 1km), 0.290 (0.230 for 1km) and 0.201 (0.216 for 1km), respectively, compared to 0.85 (0.70 for 1km), 0.52 (0.65 for 1km) and 0.45 (0.55 for 1km) for our study (see Table 1). Russell et al. (2019) found normalized mean biases for $O_3$, $SO_2$, and $PM_{2.5}$ to be 52.7% (55.9% for 1km), 113% (137.6% for 1km) and -26.8% (-25.6% for 1km), respectively, compared to our NMB of 27.3% (15.9% for 1km), 224% (1061% for 1km) and 26.3% (81.4% for 1km).

Comparing against a similar study by Stroud et al. (2020) that examined short-term GEM-MACH performance over Toronto,

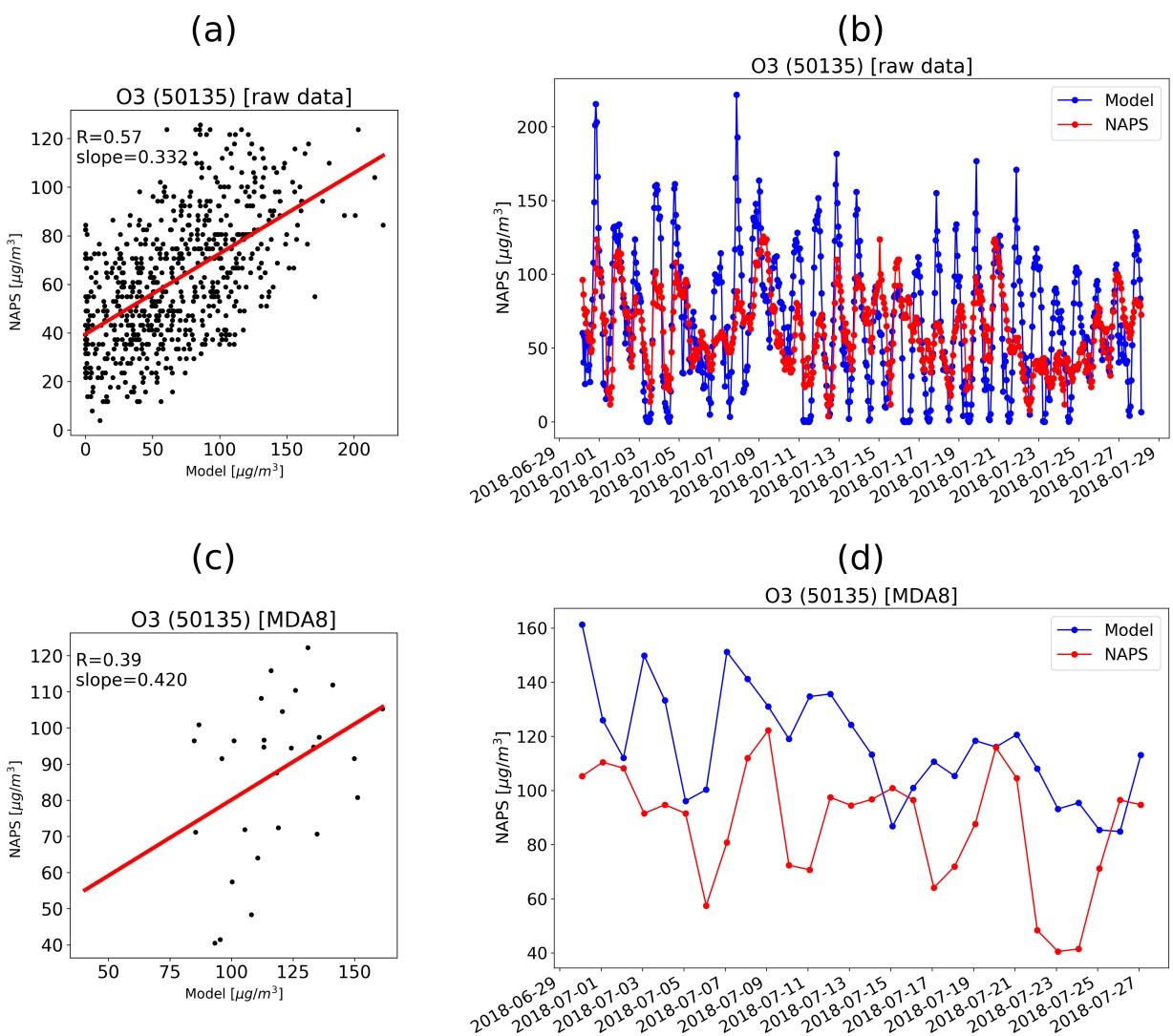

**Figure 6.** Correlation ((a) for raw data, and (c) for MDA8) and time series ((b) for raw data, and (d) for MDA8) of NAPS data and the modeled $O_3$ over a NAPS site in Montreal (NAPS ID: 50135) in July. All units are in $\mu g/m^3$.

Ontario, Canada for $O_3$ and $NO_2$ at both 10km and 2.5km resolution using NAPS surface observations, also show comparable correlations. While their modeling duration was much shorter (limited to several days in July 2015) and thus a direct comparison could not be made, their study saw correlation coefficients of 0.62 and 0.77 for $O_3$ and $NO_2$, respectively, compared to 0.85 and 0.70 for our 3km resolution runs. Similar to the comparison with Russell et al. (2019), our model showed higher biases; they saw $O_3$ normalized mean biases of 5.4% with their 2.5km resolution run, compared to 21.4% for our 3km run, and 28.2% for $NO_2$ compared to 78% for our 3km run.

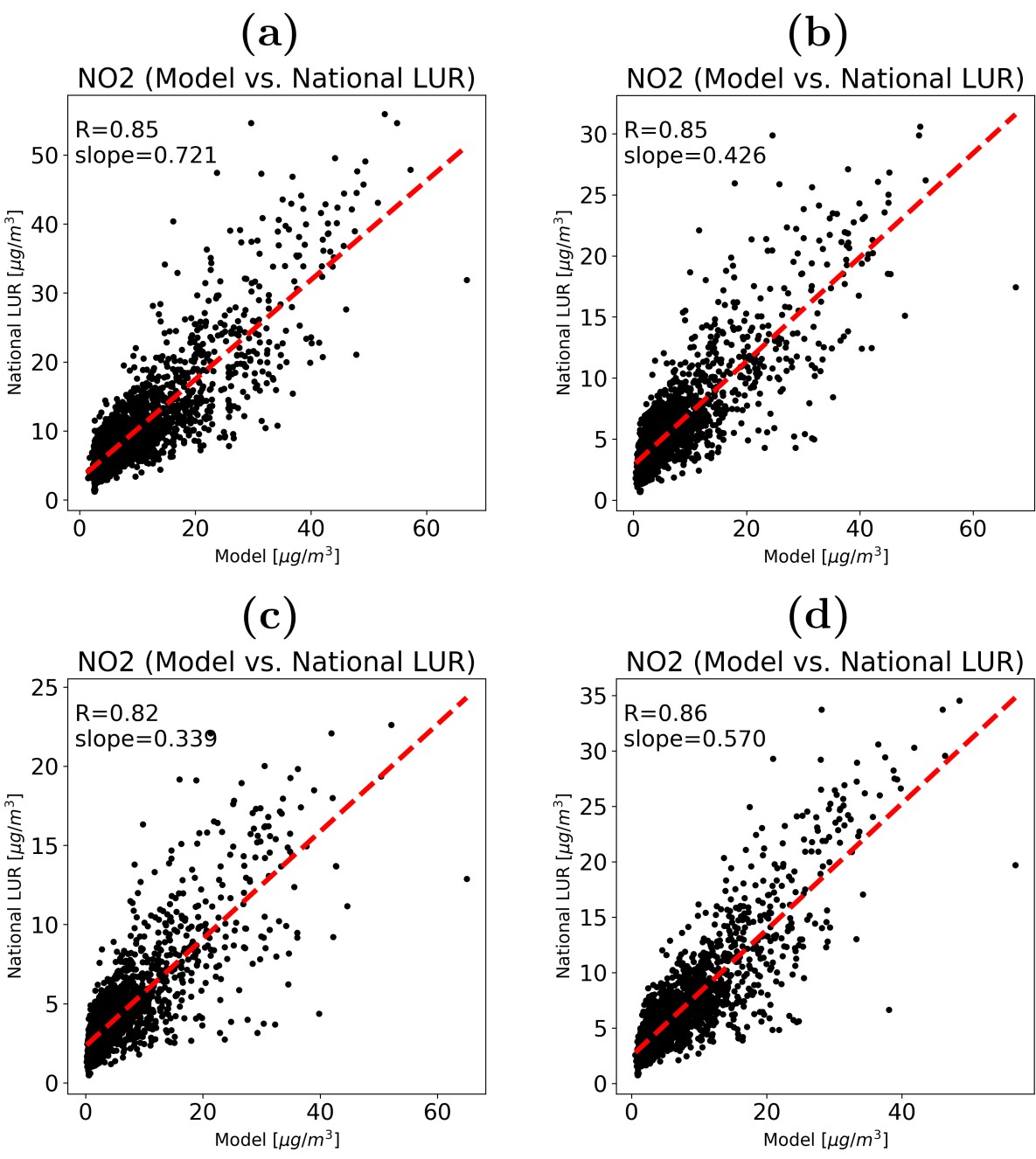

**Figure 7.** Monthly model vs. CANUE National LUR NO$_2$ correlation plots for (a) January, (b) April, (c) July and (d) October. All units are in μg/m$^3$.

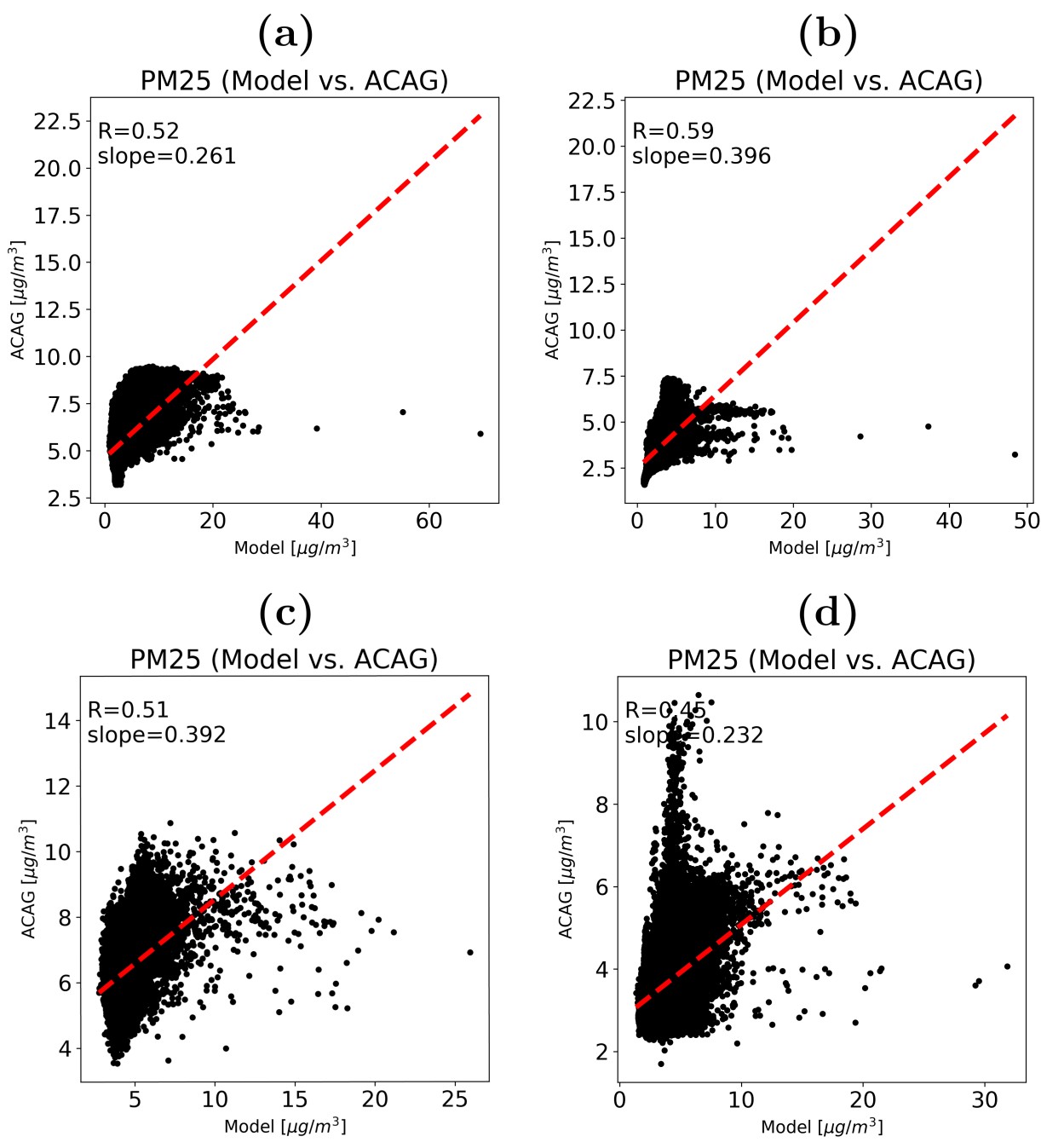

**Figure 8.** Monthly model vs. ACAG PM$_{2.5}$ correlation plots for (a) January, (b) April, (c) July and (d) October. All units are in μg/m$^3$.

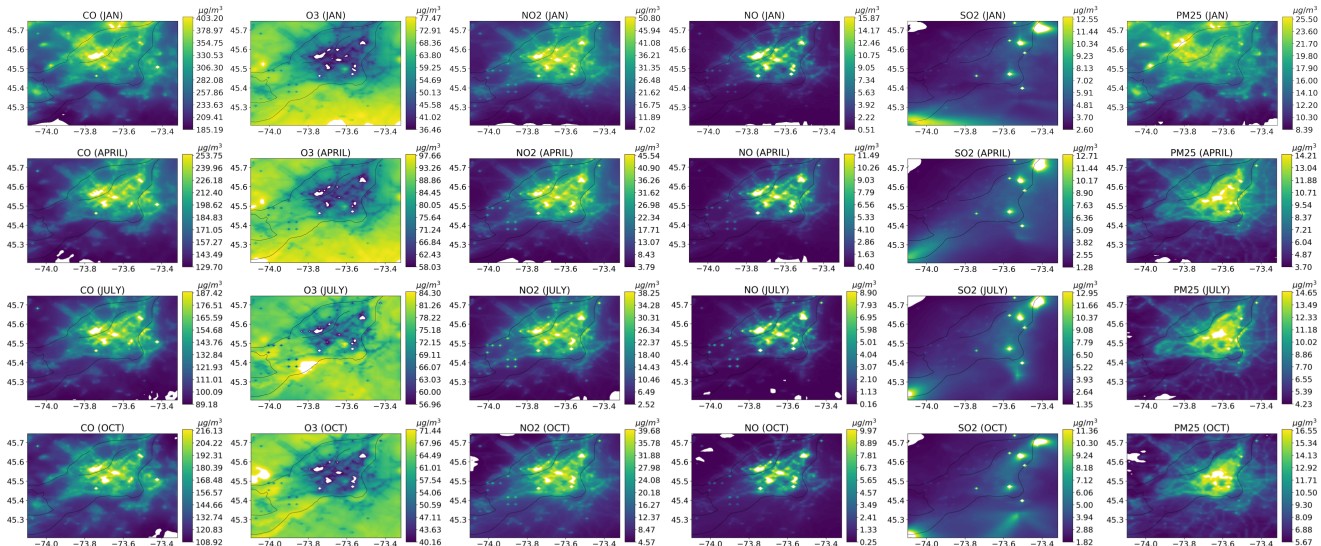

**Figure 9.** Monthly averages (January, April, July and October, from top to bottom) of the Polair3D model at the 1km resolution over Montreal for (from left to right) CO, $O_3$, $NO_2$, NO, $SO_2$ and $PM_{2.5}$. All units are in $\mu g/m^3$.

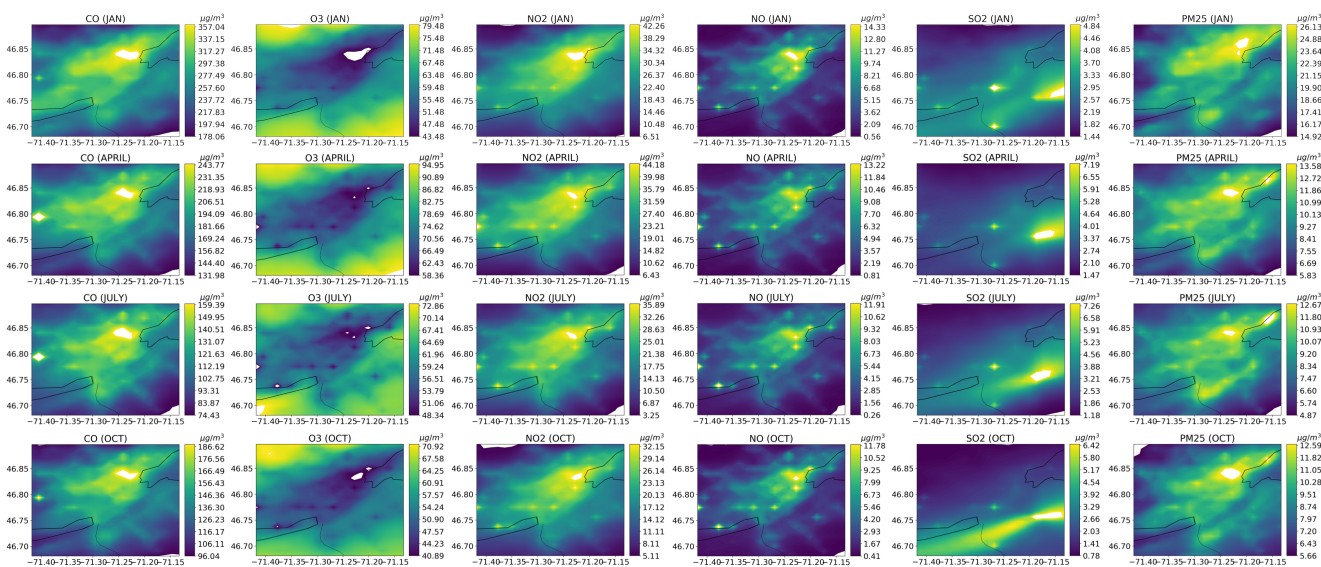

**Figure 10.** Monthly averages (January, April, July and October, from top to bottom) of the Polair3D model at the 1km resolution over Quebec City for (from left to right) CO, $O_3$, $NO_2$, NO, $SO_2$ and $PM_{2.5}$. All units are in $\mu g/m^3$.

Our Polair3D runs can also be compared to other studies that used the same model over Europe. Lugon et al. (2020) found in their study that Polair3D, over Paris, France at 1km resolution, underestimated $NO_2$, while our study overestimated it. A

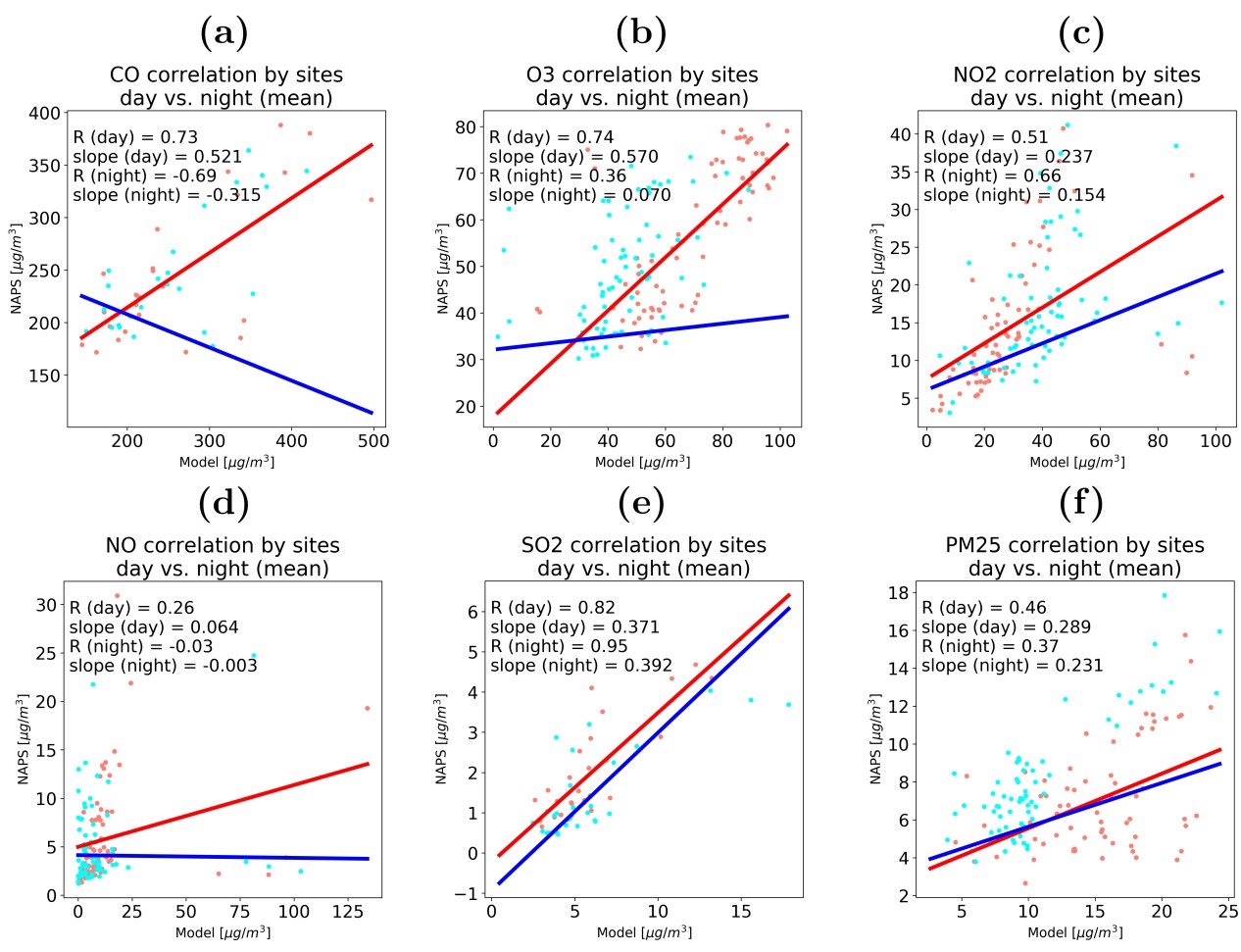

**Figure 11.** Monthly average site-wide correlation plots, separated between day (red) and night (blue) data points, at the 1km resolution for (a) CO, (b) $O_3$, (c) $NO_2$, (d) NO, (e) $SO_2$ and (f) $PM_{2.5}$. All units are in $\mu g/m^3$.

large-scale (spanning all of Europe), low resolution (0.5° by 0.5°) and long-term (2000-2008) Polair3D study by Lecœur and Seigneur (2013) reported correlation coefficients of 0.629 and 0.591 for $O_3$ and $PM_{2.5}$, respectively, comparable to our 3km

270  values (0.85 and 0.45 for $O_3$ and $PM_{2.5}$, respectively).

### 3.4  Test Scenarios: Effects of Industrial Emissions

To assess the model behavior under varying emissions scenarios, a run with no industrial emissions was performed for the same domain and time frames. Monthly averages plots showing "full emissions" (regular) run subtracted from "no-industrial-emissions" run are shown in Figures 13, 14 and 15, for 3km, Montreal 1km and Quebec City 1km runs, respectively. Here,

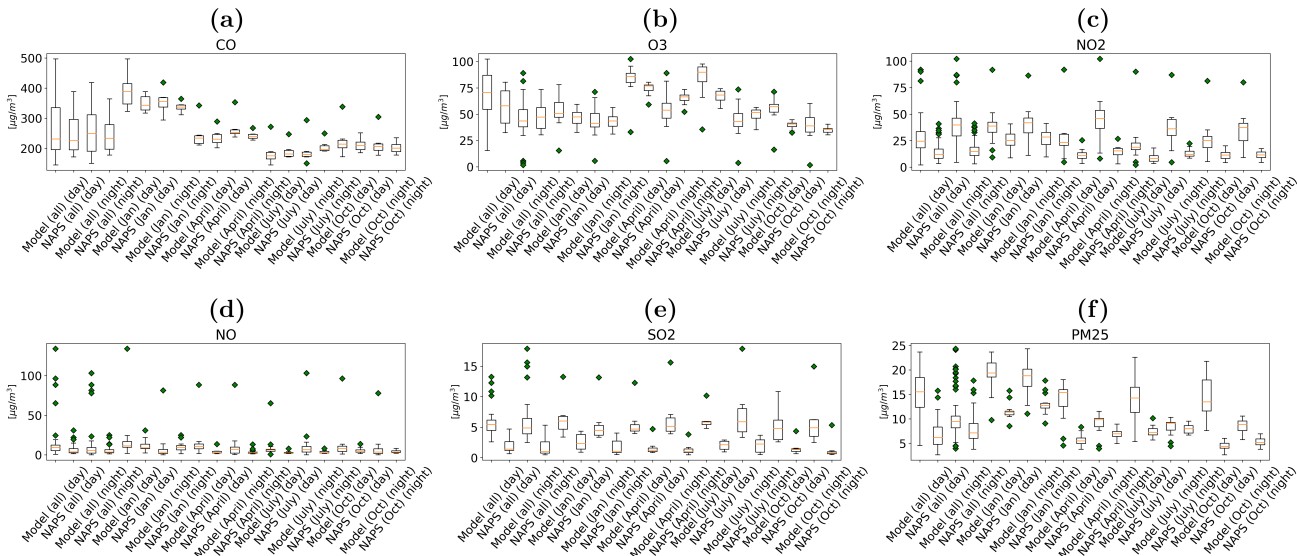

**Figure 12.** Monthly average site-wide box plots, separated between day and night data points, at the 1km resolution for (a) CO, (b) $O_3$, (c) $NO_2$, (d) NO, (e) $SO_2$ and (f) $PM_{2.5}$. All units are in $\mu g/m^3$.

275    the negative values (the blue areas) indicate industrial emission hotspots, showing the local influences of these industrial sites. Locations corresponding to major industrial emitters (sites with the top 20% and top 50% emissions for the 3km and 1km runs, respectively) from the NPRI inventory are shown in the same figures; in these plots, the black X's indicate NPRI emissions corresponding to each species, except for $O_3$, NO and $NO_2$, where $NO_x$ emission sites were used. In both 3km and 1km runs, $O_3$ showed strong seasonal variation due to the photochemical nature of its creation and destruction.

280     The model generally captures the spatial variability of industrial emissions, and captures spatial gradients related to proximity to industrial sources. Industrial contribution is not high overall (e.g., compare Figures 3 and 13). However, large spatial gradients in emissions contributions are clearly visible, for example over Montreal (Figure 14) where clear hotspots can be distinguished even at the intracity level, and because of the large spatial gradients, it remains a concern for health issues in certain areas of Quebec with high industrial activities.

285     The test run with the smelter and refinery industry suppressed showed reductions mainly centered around Trois-Rivières. This can be seen in Figure 16. Trois-Rivières is Canada's oldest industrial city. Comparing Figures 13 and 16 shows that CO contribution from smelter and refinery industries are substantial. The scenario with only the emissions from paper and pulp industry turned off showed similar results. Figure 17 shows the deltas in monthly averaged concentrations. Paper and pulp deltas were highest around Thurso, Sherbrooke, as well as Trois-Rivières.

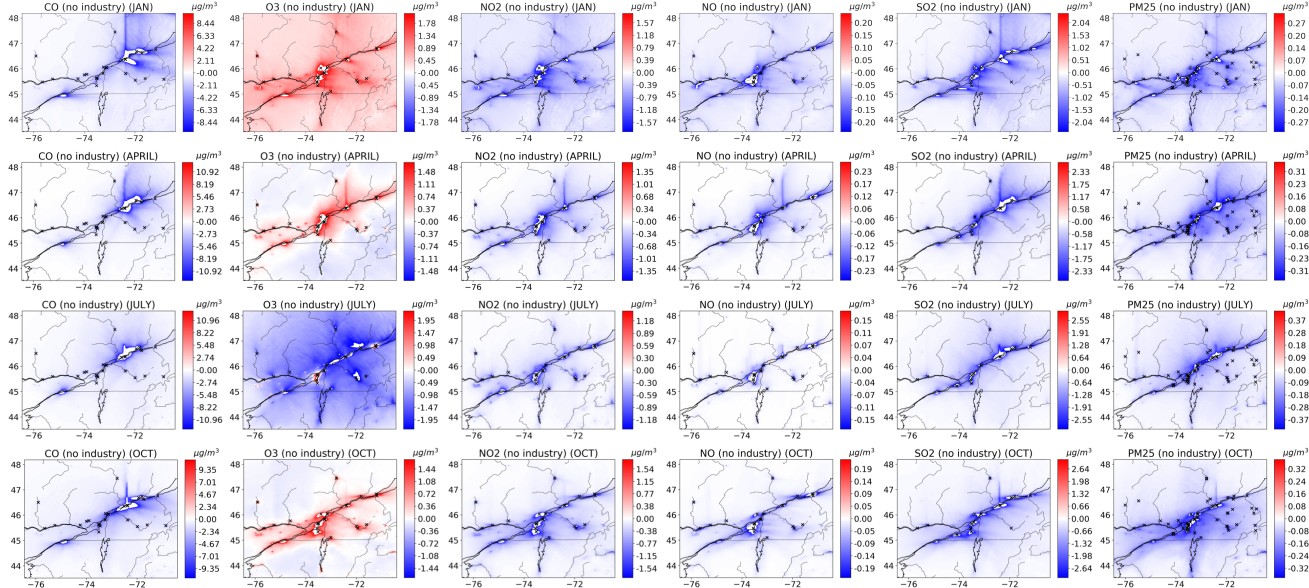

**Figure 13.** Monthly averaged plots (January, April, July and October, from top to bottom) showing the model with full emissions subtracted from a run with no industrial emissions, at the 3km resolution for (from left to right) CO, $O_3$, $NO_2$, NO, $SO_2$ and $PM_{2.5}$. The X's indicate NPRI emissions corresponding to each species, except for $O_3$, NO and $NO_2$, where $NO_x$ emission sites were used. All units are in $\mu g/m^3$.

## 4   Conclusions

In this study, the Polyphemus Polair3D CTM was run over Quebec, Canada to assess the model's capability in predicting key air pollutant species over the region at the ground (surface) level model, at seasonal temporal scales and at regional spatial scales. This represents a novel use of the Polair3D model; this study presents, the best of our knowledge, the first time the Polair3D model was used over Quebec, Canada with a long enough modeling period to capture seasonal effects, and a large modeling domain spanning urban to rural areas.

The model was run in 3 nested domains; the largest and coarsest-resolution domain was roughly 9km by 9km grid-cell resolution (edges), and within it, a smaller 3km by 3km resolution was run, and lastly, a 1km by 1km resolution runs were performed over Montreal and Quebec City. The model was run with meteorology field from pre-run WRF, and SMOKE emissions-processing system was used to prepare the emissions input files. Canadian and United States (US) emissions in the domain were calculated based on SMOKE-ready formats of the Canadian emission inventory and US National Emission Inventory, along with their temporal allocation and chemical speciation data. Spatial allocations for the three nested domains were generated using both Canadian and US spatial allocator inputs. The model was run for four seasons out of 2018, with four weeks per season (January for winter, April for spring, July for summer, and October for fall), for a total of 16 weeks of model data. Spin-up was done for 1 week for each run. Boundary conditions for the outermost domain, and the initial conditions for each of the runs were derived from CAM-Chem assimilated data.

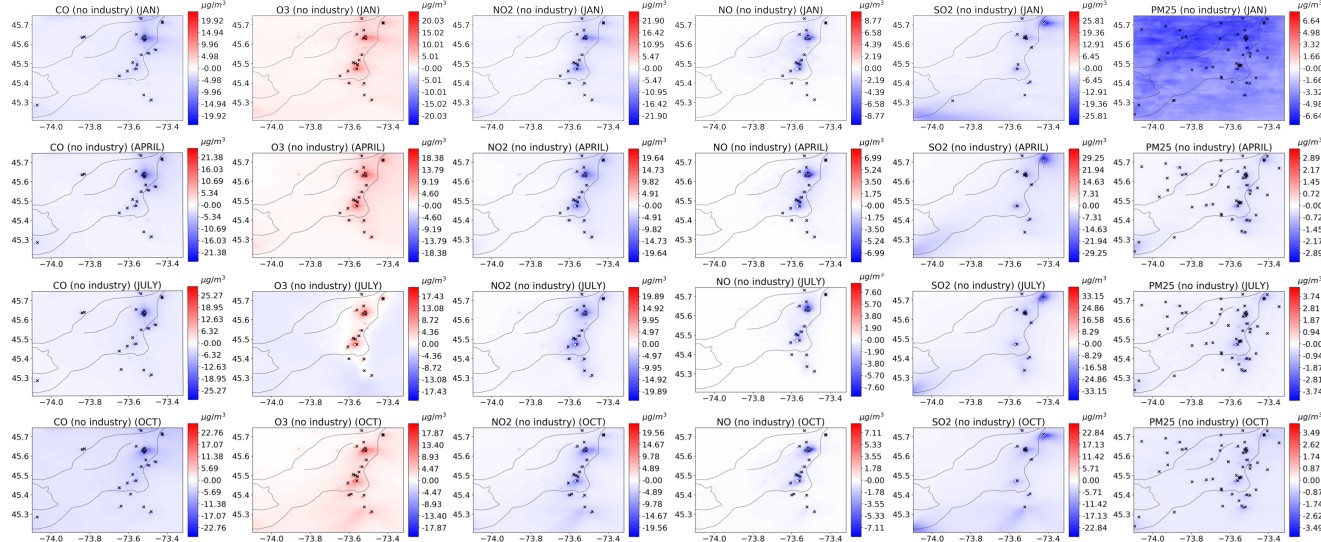

**Figure 14.** Monthly averaged plots (January, April, July and October, from top to bottom) showing the model with full emissions subtracted from a run with no industrial emissions, at the 1km resolution over Montreal for (from left to right) CO, $O_3$, $NO_2$, NO, $SO_2$ and $PM_{2.5}$. The X's indicate NPRI emissions corresponding to each species, except for $O_3$, NO and $NO_2$, where $NO_x$ emission sites were used. All units are in $\mu g/m^3$.

The model at the 3km resolution showed varying levels of performance for different pollutant species. The model at both the 3km and the 1km resolution struggled to capture high frequency temporal variability, at least at the surface, and showed large variabilities in correlation and bias from site to site. Doing a site-wide analysis (i.e., comparing monthly averages across all sites) suggested that the model is better at capturing the spatial variability and seasonal effects, rather than hour-by-hour or day-to-day temporal variability for a fixed location.

When comparing the biases and correlation at a site-wide scale, the higher resolution 1km runs did not result in strictly better performance; when analyzing the same sites (i.e., restricting the 3km analysis to the NAPS sites seen in the smaller 1km run), the 3km model showed slightly higher correlation for $O_3$, $NO_2$, and NO and while $SO_2$ and $PM_{2.5}$ showed increases in correlation, the difference were not large (CO correlation was the same for both cases). Examining the model performance site by site showed similar results; Running the model at an increased resolution may be an effective way to downscale the data, but it does not appear to make the simulation more temporally accurate. Comparing against CANUE National LUR $NO_2$ showed high correlations, ranging between $R = 0.91$ (in July) and 0.86 (in October). At the 1km resolution, another analysis was conducted, separating day and night time data; one noteworthy result from this analysis is that the slope was higher during the day for all species except $SO_2$. Correlation was higher for CO, $O_3$, NO and $PM_{2.5}$, and was slightly lower for $NO_2$ and $SO_2$. Furthermore, $O_3$, a secondary pollutant that is created and destroyed photochemically and thus heavily affected by sunlight, showed higher correlation during the day than night, while and at the same time showed large underestimation biases during the night. This suggests that the model is capable of modeling $O_3$ during the day but struggles to simulate the background $O_3$

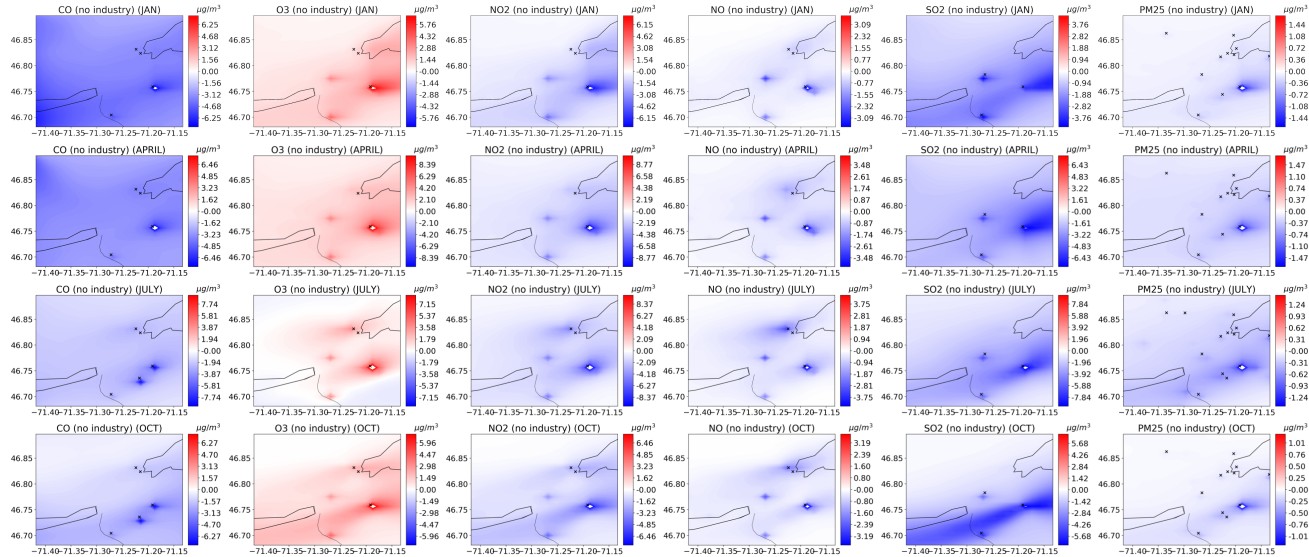

**Figure 15.** Monthly averaged plots (January, April, July and October, from top to bottom) showing the model with full emissions subtracted from a run with no industrial emissions, at the 1km resolution over Quebec City for (from left to right) CO, $O_3$, $NO_2$, NO, $SO_2$ and $PM_{2.5}$. The X's indicate NPRI emissions corresponding to each species, except for $O_3$, NO and $NO_2$, where $NO_x$ emission sites were used. All units are in $\mu g/m^3$.

during the night where photochemical reactions are low and/or nonexistent. For CO, correlation was significantly higher during the day than night ($R = 0.91$ versus -0.69), although the difference was less extreme when looking at individual months.

A test scenario, where the model was run without industrial emissions, showed that the model generally captures the spatial variability of industrial emissions, and captures spatial gradients related to proximity to industrial sources. While industrial contribution is not high overall, large spatial gradients were seen in its contributions, even at intracity scales.

The performance of the Polair3D model over Quebec was in line with other models like GEM-MACH over Canada albeit with higher biases overall, and comparable to the performance of Polair3D over Europe, where the model was developed. For

key air pollutants such as $O_3$ and $NO_2$, Polair3D showed similar correlations and comparable biases.

*Code availability.* The Polyphemus platform, including the Polair3D CTM used in this study, is available at https://doi.org/10.5281/zenodo. 10067062 (Kim et al., 2023).

*Author contributions.* SY ran the model with emissions inventories prepared by SMG. All authors contributed valuable discussions to this study. SY wrote the paper with input from all authors.

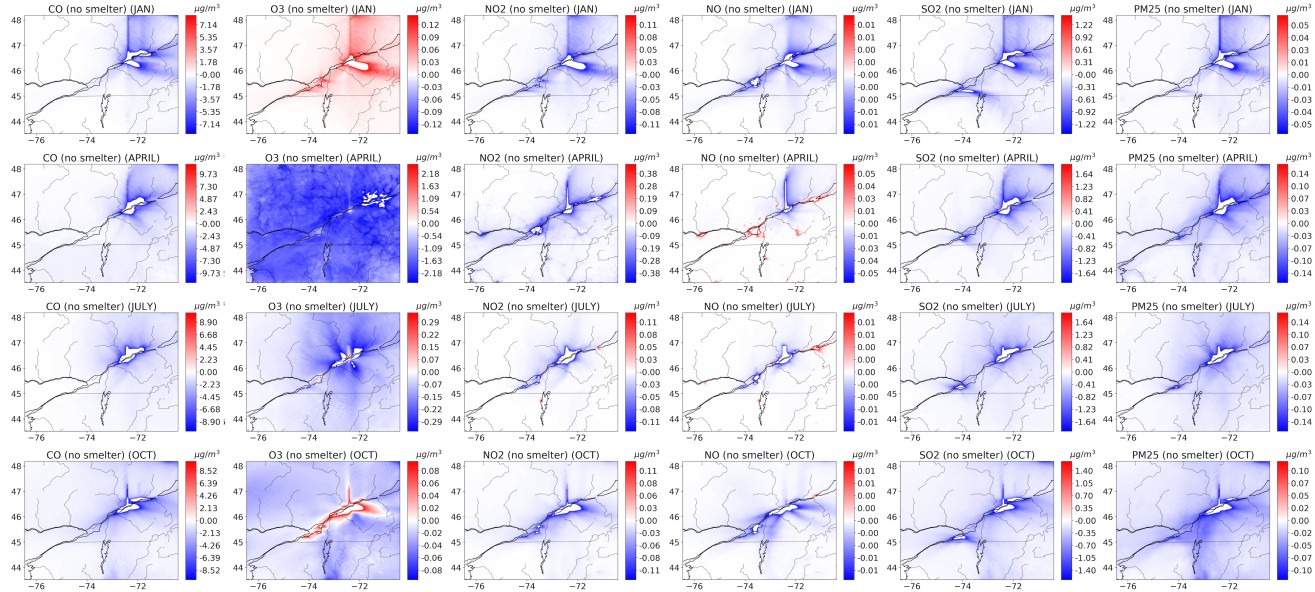

**Figure 16.** Monthly averaged plots (January, April, July and October, from top to bottom) showing the model with full emissions subtracted from a run with no emissions from smelter and refinery industries, at the 3km resolution for (from left to right) CO, $O_3$, $NO_2$, NO, $SO_2$ and $PM_{2.5}$. All units are in $\mu g/m^3$.

*Competing interests.* The contact author has declared that none of the authors has any competing interests.

*Acknowledgements.* This work was funded by Health Canada's Addressing Air Pollution Horizontal Initiative.

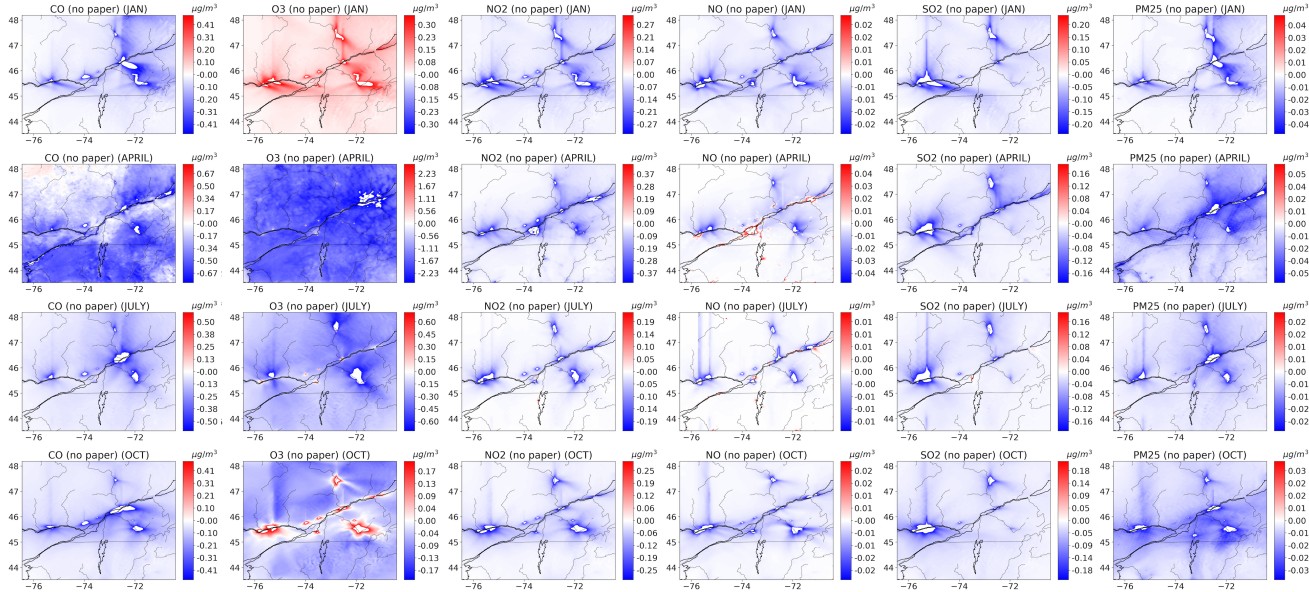

**Figure 17.** Monthly averaged plots (January, April, July and October, from top to bottom) showing the model with full emissions subtracted from a run with no emissions from paper and pulp industries, at the 3km resolution for (from left to right) CO, $O_3$, $NO_2$, NO, $SO_2$ and $PM_{2.5}$. All units are in $\mu g/m^3$.

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
