# Peer review of "Validation and Analysis of the Polair3D v1.11 Chemical Transport Model Over Quebec"

_EGUsphere, 2023_

## Author Comment (AC1)

egusphere-2023-2038
Response to Reviewer Comment 1

We thank the reviewer for the constructive comments. This document summarizes our responses and documents the changes made to the manuscript.

All reviewer comments are written in black font
All responses are written in red font
All references to changes in the revised manuscript are written in **bold red font highlighted P**

*The paper's focus is on the validation of the Polair3D chemistry transport model over the geographical domain of Quebec, Canada at different spatial and temporal resolutions. The authors highlight that the model was used for the first time over the Quebec domain and for the first time with a temporal period long enough to evaluate its performance at the seasonal level. The authors present a statistical analysis of the model performance in representing a range of primary and secondary air pollutants comparing model results with ground observation sites. Finally, they conduct a sensitivity test on the primary emissions impact on final concentrations switching off the industrial emissions and evaluating the change in the levels of air pollutants. The manuscript is, on the one hand, highly focused on the evaluation of the model performance in representing a range of pollutants on a large temporal scale. This represents a challenge for any CTM because of the impact that meteorology and chemical mechanisms can have at seasonal levels and at different resolutions. On the other hand, there is a low focus on the expendability of the model (E.g., scenario analysis). The manuscript would benefit from a clearer statement of the use the authors want to do of the model. They mention at the beginning the impact that industrial air pollution has on human health, and they create a scenario to test the model's performance. Nevertheless, there is no health impact quantification analysis of model outputs or scenarios.*

*A more careful choice of the use of the model would give higher focus to the validation and the choice and description of proposed scenarios. If the choice of the model is motivated by "scenarios-impact" analysis then the validation should focus on high-resolution simulations and on a temporal scale that would allow to evaluate the model against national or international threshold limits at a daily/hourly level. Contrarywise, if the intention is to use the model for monthly regional simulations then the validation could be limited to a 3x3km resolution and evaluate the model representation of seasonal average values analysing the impact of scenarios at the annual/seasonal level.*

**Major Comments:**
***43 – 46:*** *The authors apply the Polair3D model over the domain of Canada highlighting that this particular model has seen only a little use over North America and Canada. It should be made clearer why the use of this particular model should represent a step ahead in air pollution research. Several types of CTMs serve for different purposes from different points of view. These can be related - for example - to the representation (or absence) of particular chemical*

*mechanisms to describe the chemical life cycle of some pollutants, or to minor computational costs that make the simulations quicker or smaller in terms of storage space. The authors should make clearer and stronger the motivations that led them to 1) choose the Polair3D model and 2) which use it besides the pure capability of representing concentrations of air pollutants (e.g., scenarios, forecasts, mitigation policies testing, transport/trajectories analysis).*

While this paper presents model set-up and validation, the ultimate purpose of this air quality modeling exercise is to support the analysis of sector-specific mitigation policies and their benefits from a population health perspective. This entails the ability to run multiple scenarios and the need for high resolution modeling especially in populated areas. POLAIR3D is part of a suite of air quality models, the POLYPHEMUS platform, and lends itself for plume-in-grid and street-in-grid applications. While this paper does not present either application given that the scope is limited to validation and performance assessment of the CTM, our ultimate goal is to capture the spatial variability of concentrations under mitigation scenarios, to support epidemiologic analyses.

In Canada, two other CTM platforms are typically used: the USEPA's Community Multiscale Air Quality Modeling System (CMAQ) commonly used for studies in North America, and GEM-MACH run by ECCC. CMAQ's major difference with Polair3D lies in its management of the aerosols, while the aerosol module in Polair3D is a sectional model, CMAQ is a modal model. Furthermore, CMAQ has mainly been applied at regional scales, i.e., using horizontal spatial resolutions of 36, 12, or 4 km², with few studies at local scales, i.e., with a resolution of 1 km². In contrast, Polair3D has been widely used with horizontal resolutions of 1 km² with robust performance. The ECCC's GEM-MACH model is currently unavailable for academic use and is not set-up for the type of resolution and scenario analysis required for this project. Several studies have shown that CTMs simulating aerosols for exposure and health assessment analyses over urban areas should be conducted at the finest spatial resolution possible, since higher resolution would reduce exposure misclassification for the population, and enhance the accuracy of health risk estimates.

==Text summarizing this was added to the manuscript.==

For the benefit of the reviewer, we provide below a table summarizing the sub-modules in Polair3D, CMAQ, and GEM-MACH.

| Table 1. Comparison of the sub-modules in Polair3D, CMAQ, and GEM-MACH | | | |
|---|---|---|---|
| | POLAIR3D | GEM-MACH | CMAQ |

| Model | Polair3D | Global Environmental Multiscale-Modelling Air-quality and Chemistry (GEM-MACH) | Community Multiscale Air Quality Modeling System (CMAQ) |
|---|---|---|---|
| Type | Eulerian CTM | Online model (meteorology and chemistry are handled within a single model | Eulerian CTM |
| Meteorology | WRF, MM5 | Includes a "physics and chemistry processor" (GEM physics module) | WRF data |
| Initial/boundary conditions | MOZART, CAM-Chem, GEMS etc. CAM-Chem recommended for recent dates (CAM-Chem is available for download online) | GEM forecast for boundary conditions | CMAQ Chemistry Transport Model (CCTM),GEMS |
| Aerosol | Choice of several aerosol modules; SOAP (Secondary Organic Aerosol Processor), SCRAM (new, recommended), SORGAM (requires ISORROPIA)

SIREAM is a sectional or size-resolved model (the aerosol size distribution is represented by a number of sections) | Model includes: sedimentation, nucleation, condensation, coagulation, swelling, activation, sea-salt emissions, and inorganic gas-particle partitioning, as well as SOA formation

Aerosol capability is based on CAM | (AERO7) is introduced in CMAQv5.3; Aerosol module uses ISORROPIA v2.2 in the reverse mode to calculate the condensation/evaporation of volatile inorganic gases to/from the gas-phase concentrations of known coarse particle surfaces

CMAQ is a modal model (log-normal distributions,

or modes, are superposed to represent the aerosol size distribution) |
| Chemistry | CB05 (52 species, 155 reactions), RACM (72 species, 237 reactions), | ISORROPIA (heterogenous chemistry) ADOM/ADOM-II | ISORROPIA, and separate aqueous chemistry module. Aqueous chemistry and scavenging is calculated for resolved clouds as well, using the cell liquid water |

| | | | |
|---|---|---|---|
| | RACM2 (113 species, 349 reactions) | (aqueous and gas phase chemistry) | content and precipitation from the meteorological model. |
| Speci es | 52 species (with CB05 chemistry module) | Aerosols: PM (2.5 and 10), chemical breakdown (SO4, NO3, NH4, EC, pOC, sOC, CM, SS)

Chemical species: O3, NO2 | Listed here: https://github.com/USEPA/CMAQ/blob/main/DOCS/Users_Guide/CMAQ_UG_ch06_model_configuration_options.md#6.11_Aerosol_Dynamics |
| Notes | Polair3D is a Eulerian CTM, and can be coupled with chemistry/aerosol modules e.g., SIREAM

Polair3D has been widely used with horizontal resolutions of 1 sq km with robust performance | Developed for AQ forecasting (O3, NO2 and PM); developed by and for Canadian air quality analysis | Relies on the open source Sparse Matrix Operator Kernel Emissions (SMOKE) model to estimate the magnitude and location of pollution sources

CMAQ has mainly been applied at regional scales(i.e., using horizontal spatial resolutions of 36, 12, or 4 sq km, with few studies at local scales, i.e., with a resolution of

1 sq km) |
| Refer ences | http://cerea.enpc.fr/polyphemus/doc/Polyphemus-1.11-Guide.pdf

https://link.springer.com/article/10.1007/s11869-019-00733-5/tables/1 | https://collaboration.cmc.ec.gc.ca/science/rpn/SEM/dossiers/2009/seminaires/2009-05-08/Seminar_2009-05-08_Donald-Talbot.pdf

https://link.springer.com/article/10.1007/s11869-019-00733-5/tables/1

https://doi.org/10.5194/gmd-11-2609-2018

https://slideplayer.com/slide/6540524/ | https://github.com/USEPA/CMAQ/tree/main/DOCS/Users_Guide

https://doi.org/10.5281/zenodo.5213949 |

1. Solazzo, Efisio, et al. "Ensemble modelling of surface-level ozone in Europe and North America for AQMEII." *Air Pollution Modeling and its Application XXII*. Springer, Dordrecht, 2014. 351-356.
2. Silveira, Carlos, Joana Ferreira, and Ana Isabel Miranda. "The challenges of air quality modelling when crossing multiple spatial scales." *Air Quality, Atmosphere & Health* 12.9 (2019): 1003-1017.

*61 – 64: The performance of a CTM is highly influenced by the levels of primary emissions and by the representation of the regional meteorology. The authors mention that the meteorology used to drive the model in representing the air pollution was taken by WRF, but they don't mention a validation of this meteorology. Analysing the performance of the model at the seasonal level and focusing on primary and secondary pollutants it would be good to understand the levels of reliability of parameters such as surface temperature, solar radiation, wind speed and temperature, and relative humidity. The evaluation of these parameters should be also analysed in the same temporal dimension (e.g., seasonal, annual) of the air pollution concentrations.*

As meteorological data largely affect model results, we examined the meteorological input files from the Weather Research and Forecasting (WRF) model, and validated them against observational temperature and wind data. This validation was done at all spatial resolutions, at hourly temporal resolution. Correlations, root mean square error (RMSE) and biases were calculated for temperature, wind speed, and wind directions. This was done at all meteorological stations in our modeling domain: those operated by Environment and Climate Change Canada (ECCC) and provinces. We have updated the manuscript and added more information regarding WRF configurations.

As examples, Figures 1, and 2 present a comparison of WRF outputs and observational data at Montreal Trudeau airport at 1km and 3km resolutions. WRF output for October 1 to 14 2018 were compared against observations. Data were collected by ECCC and were downloaded from their publicly available website.

[Figure]

Figure 1. Time series plots of model (at 1km resolution) and in-situ observations of temperature (top left), zonal wind (bottom left), and meridional wind (bottom right) seen at Montreal P.E. Trudeau airport. Temperature correlation plot is also shown (top right). In all the time series plots, observation is indicated by blue, model by red, and the deltas by dashed grey.

[Figure]

Figure 2. Time series plots of model (at 3km resolution) and in-situ observations of temperature (top left), zonal wind (bottom left), and meridional wind (bottom right) seen at Montreal P.E. Trudeau airport. Temperature correlation plot is also shown (top right). In all the time series plots, observation is indicated by blue, model by red, and the deltas by dashed grey.

*92 – 93: Anthropogenic emissions come from NEI 2014 and EPA 2017 inventories. The authors mention using the SMOKE pre-processor and a combination of the "SMOKE-ready" format of these inventories. Are these the most up-to-date inventories to represent the emissions in Quebec? The authors should mention how these two inventories have been processed and speciated before being merged. Is it assumed that – individually – the two inventories have been processed in SMOKE and then merged? If yes, in which way and according to which criteria? Besides the representation of the PM, how the VOCs are represented/speciated in the model? The choice of anthropogenic emissions can be critical in the representation of the final concentrations, and it would be good to have a quantitative analysis (even only in the supplementary material) that shows the annual totals. This is also in light of the scenarios with reduced emissions from Industrial sources. It's good to have maps showing their positions but it would be good to know which percentage of reduction this sector, once deleted, gives on the totals.*

**Canadian inventory estimates are given in detail in a report by ECCC. We refer to this report in the paper, including a briefl excerpt of annual total emissions.** We did not include figures in the SI simply because they already exist in the published report, even at a provincial level. https://www.canada.ca/en/environment-climate-change/services/air-pollution/publications/emissions-inventory-report-2022.html

This study uses two emission inventories, the Canadian inventory and the US inventory.

Canada's Air Pollutant Emissions Inventory (APEI) also known as Canadian criteria-air-contaminants (CAC) emissions inventory, prepared and published by Environment and Climate Change Canada (ECCC). The APEI is a comprehensive inventory of anthropogenic emissions of 17 air pollutants such as $CO$, $NH_3$, $NO_x$, $PM_{2.5}$, $PM_{10}$, $SO_2$, and VOC at the national, provincial, and territorial levels and it is compiled from many different data sources. This is the most up to date complete inventory for Canada.

Since the studied domain includes some regions of the United States, the US inventory, also known as the National Emission Inventory (NEI) was processed as well. A combination of SMOKE-ready formats of NEI 2014 and 2017 inventories were taken into account.

The two inventories have been processed using SMOKE individually for each specific source. Through this process, we derived the gridded speciated hourly emission files for each source, then all the area sources were merged into one binary file representing total surface emissions. All industrial sources were merged into one binary file representing total volume emissions. Both surface emissions and volume emissions were used as inputs in the Polair3D model.

For this study, we used the Carbon Bond mechanisms (CB05) for VOC speciation and for the gas-phase mechanism, as well as the AE6 aerosol scheme.

SMOKE-ready format of the Canadian inventory for the year 2015 was used. The APEI is compiled by the Pollutant Inventories and Reporting Division (PIRD) of ECCC. The inventory databases compiled by PIRD are modified by the Air Quality Modelling Applications Section (AQMAS) of ECCC for emissions processing with SMOKE for the Canadian air quality modeling platform. These are the most up to date complete inventories. SMOKE has four individual stages for emission processing including, chemical speciation, temporal allocation, spatial allocation, growth, and control scenario definition. In this project, we consider the three stages of SMOKE emission processing that are shown in Figure 3 to obtain hourly gridded regional emissions for the modeling domain. SMOKE can process different types of emission inventories such as area sources, point sources, and mobile sources. In this study, we have processed all ECCC emission inventories as area sources.

[Figure]

Figure 3. The emissions processing scheme used in this study

*113 – 115: The authors mention the use of the NAPS observation sites for the evaluation of the model performance. It would be good to know where these observation points are (maybe in Figure 1) and most of all which type of observation sites these are. Are urban backgrounds and or rural sites? Any of these are road traffic sites? The model performance could sensibly change if different sites are computed together or by type. The suggestion here is to divide the sites by type and perform the statistical analysis again. The evaluation of urban background sites for NO, NO2 and O3 could reveal information about how the model represents titration processes in urban environments while in rural areas it could be analysed the impact that biogenic emissions of VOCs have on O3.*

An analysis evaluating model performance at different NAPS site types was added to the revised manuscript. Here, the NAPS designated classifications of regional background (RB), general population exposure (PE), and transportation-influenced (T) were used. Transportation-influenced sites showed the highest correlation for NO2, NO and O3, but also had the fewest data points. For O3, the highest correlation was seen with the RB sites, which is in line with the previous analyses that indicate that the model is able to capture the overall amount of background O3 that is generated/destroyed. For CO, there were no RB sites available, and both T and PE sites showed similar correlations.

*149 – 160: The evaluation of the performance in representing NO, NO2 and O3 would benefit from more information about VOCs, and by how the original NOX emissions are partitioned in NO and NO2. For what concern is the performance of PM2.5 Is there any transport pattern that could influence the seasonal variability in the model performance? The authors mention that the*

*model performance in summer is lower for all pollutants except for SO2. Is there any reason related to meteorology or seasonal emissions of SO2 that could give this?*

Adding to the comment above, the site type analyses sheds some light on the model performance of NOx and O3. The lack of correlation in PE NO (R=-0.06), along with the PE O3 R=0.70 and PE NO2 R=0.48 indicate that the emissions of NO in urban areas may be inaccurate, but NO2 fares better, and the model is able to still capture the overall urban O3 trends. For PM2.5, the PE sites had the highest correlation, although the correlation was still relatively low at R=0.38; the model struggles to capture the seasonal and spatial variability of PM2.5.

Across all sources, $NO_x$ emissions were subdivided into nitric oxide (NO) (90%) and $NO_2$ (10%); NO and NO2 were speciated within the SMOKE. Please refer to the following documentation: https://www.cmascenter.org/smoke/documentation/4.5/html/ch02s11.html

*178 – 182: The authors mention that the model performance does not increase with the model resolution for CO and NO2. This can be seen from the parson coefficient by going to inspect the MB in Table 1 this is always lower in the 1x1km domain. What is not mentioned in the text is that the evaluation of the "3km to 1km" shows higher MB than the "3km". This could suggest that the agreement in the model performance decreases in urban background sites (that are supposed to be denser in the 1km domain area) and is higher in rural areas. An analysis of the model performance by site type could explain this better.*

An analysis at the 3km resolution, but with sites only found in the 1km resolution domain, was done to account for this difference in site selection (see Table 1 in the revised manuscript).

*Figure 6: Why the analysis of Ozone is shown only for January when its photochemical activity is lower? Why don't show the same figure for summer and winter?*

The figures were swapped for those of July.

*Test Scenario: This part of the manuscript would benefit from a deeper understating of the impact of the reduction in industrial emissions on final concentrations and the benefit of the health impact. It's good to know where the emission points are and where the average difference in concentrations is, but it would be also good to know how much change in terms of concentrations at the receptor's sites (observation points). This would require an analysis of the meteorology to understand which sites are downwind and would benefit more from the industrial emission reduction. Additionally, the authors mention the health impact that industrial air pollution has but they don't quantify the impact that the suggested scenario could have on final concentration and health impact.*

This run section was run to simply test the model performance under varying emissions, and was not intended as a quantitative sensitivity analysis; sensitivity analysis was not the main focus of this study (that being model validation), it was included as we believed some preliminary analyses of this kind would strengthen the paper.

We have added two more scenarios on top of the "no industry" scenario (turning off emissions from the paper/pulp industry, and turning off emissions from the smelter/refinery industry) to strengthen this part of the paper.

In addition, we have performed additional analyses validating our modeled PM2.5 and NO2 against assimilated monthly ground-level PM2.5 and NO2 dataset products from ACAG (Atmospheric Composition Analysis Group) and CANUE (Canadian Urban Environmental Health Research Consortium) respectively; these datasets are widely used for air quality assessment and exposure studies. The results show fair agreements, indicating that the Polair3D model can be used for health impact analyses. Lastly, please note that we also reran the model for the month of January as we found some aberrations in some of the species in the ECCC offroad emissions (with the emissions themselves, not from our simulations). We reran the month using offroad emissions from April instead (everything else was kept the same), after checking that the offroad values did not vary significantly from month to month (in fact July and April for example were identical).

---

## Author Comment (AC2)

egusphere-2023-2038
Response to Reviewer Comment 2

We thank the reviewer for the constructive comments. This document summarizes our responses and documents the changes made to the manuscript.

All reviewer comments are written in black font
All responses are written in red font
All references to changes in the revised manuscript are written in **bold red font highlighted**

The focus of this paper is on evaluating the Polair3D CTM for a 3km horizontal grid resolution domain centered over Montreal, and two 1km resolution domains over Montreal and Quebec. Surface concentrations of CO, PM2.5, NO2, NO, SO2, and O3 from the model are shown and evaluated using ground-based observations for each season, where the model was run for four weeks per season (January, April, July, and October). A test scenario focusing on the industrial emissions is also shown.

Major Comments:

In general, the purpose of the paper is not very clear, the authors do not say why it is important that the Polair3D CTM be evaluated for Canada or why they selected that model in particular. How is applying this CTM over Quebec novel, as stated on Line 237? Has the model been used for other urban cases and at a 1km resolution, if so, what is different about Quebec (emissions, terrain, chemistry)?

While this paper presents model set-up and validation, the ultimate purpose of this air quality modeling exercise is to support the analysis of sector-specific mitigation policies and their benefits from a population health perspective. This entails the ability to run multiple scenarios and the need for high resolution modeling especially in populated areas. POLAIR3D is part of a suite of air quality models, the POLYPHEMUS platform, and lends itself for plume-in-grid and street-in-grid applications. While this paper does not present either application given that the scope is limited to validation and performance assessment of the CTM, our ultimate goal is to capture the spatial variability of concentrations under mitigation scenarios, to support epidemiologic analyses.

In Canada, two other CTM platforms are typically used: the USEPA's Community Multiscale Air Quality Modeling System (CMAQ) commonly used for studies in North America, and GEM-MACH run by ECCC. CMAQ's major difference with Polair3D lies in its management of the aerosols, while the aerosol module in Polair3D is a sectional model, CMAQ is a modal model. Furthermore, CMAQ has mainly been applied at regional scales, i.e., using horizontal spatial resolutions of 36, 12, or 4 km², with few studies at local scales, i.e., with a resolution of 1 km². In contrast, Polair3D has been widely used with horizontal resolutions of 1 km² with robust performance. The ECCC's GEM-MACH model is currently unavailable for academic use and is not set-up for the type of resolution and scenario analysis required for this project. Several

studies have shown that CTMs simulating aerosols for exposure and health assessment analyses over urban areas should be conducted at the finest spatial resolution possible, since higher resolution would reduce exposure misclassification for the population, and enhance the accuracy of health risk estimates.

Text summarizing this was added to the manuscript.

For the benefit of the reviewer, we provide below a table summarizing the sub-modules in Polair3D, CMAQ, and GEM-MACH.

| Table 1. Comparison of the sub-modules in Polair3D, CMAQ, and GEM-MACH | | | |
|---|---|---|---|
| | POLAIR3D | GEM-MACH | CMAQ |
| Model | Polair3D | Global Environmental Multiscale-Modelling Air-quality and Chemistry (GEM-MACH) | Community Multiscale Air Quality Modeling System (CMAQ) |
| Type | Eulerian CTM | Online model (meteorology and chemistry are handled within a single model | Eulerian CTM |
| Meteorology | WRF, MM5 | Includes a "physics and chemistry processor" (GEM physics module) | WRF data |
| Initial /boundary conditions | MOZART, CAM-Chem, GEMS etc. CAM-Chem recommended for recent dates (CAM-Chem is available for download online) | GEM forecast for boundary conditions | CMAQ Chemistry Transport Model (CCTM),GEMS |

| | | | |
|---|---|---|---|
| Aerosol | Choice of several aerosol modules; SOAP (Secondary Organic Aerosol Processor), SCRAM (new, recommended), SORGAM (requires ISORROPIA)

SIREAM is a sectional or size-resolved model (the aerosol size distribution is represented by a number of sections) | Model includes: sedimentation, nucleation, condensation, coagulation, swelling, activation, sea-salt emissions, and inorganic gas-particle partitioning, as well as SOA formation

Aerosol capability is based on CAM | (AERO7) is introduced in CMAQv5.3; Aerosol module uses ISORROPIA v2.2 in the reverse mode to calculate the condensation/evaporation of volatile inorganic gases to/from the gas-phase concentrations of known coarse particle surfaces

CMAQ is a modal model (log-normal distributions,

or modes, are superposed to represent the aerosol size distribution) |
| Chemistry | CB05 (52 species, 155 reactions), RACM (72 species, 237 reactions), RACM2 (113 species, 349 reactions) | ISORROPIA (heterogenous chemistry) ADOM/ADOM-II (aqueous and gas phase chemistry) | ISORROPIA, and separate aqueous chemistry module. Aqueous chemistry and scavenging is calculated for resolved clouds as well, using the cell liquid water content and precipitation from the meteorological model. |
| Species | 52 species (with CB05 chemistry module) | Aerosols: PM (2.5 and 10), chemical breakdown (SO4, NO3, NH4, EC, pOC, sOC, CM, SS)

Chemical species: O3, NO2 | Listed here:
https://github.com/USEPA/CMAQ/blob/main/DOCS/Users_Guide/CMAQ_UG_ch06_model_configuration_options.md#6.11_Aerosol_Dynamics |
| Notes | Polair3D is a Eulerian CTM, and can be coupled with chemistry/aerosol modules e.g., SIREAM

Polair3D has been widely used with horizontal resolutions of 1 sq | Developed for AQ forecasting (O3, NO2 and PM); developed by and for Canadian air quality analysis | Relies on the open source Sparse Matrix Operator Kernel Emissions (SMOKE) model to estimate the magnitude and location of pollution sources

CMAQ has mainly been applied at regional scales(i.e., using horizontal spatial resolutions of 36, 12, or 4 sq km, with few studies at local scales, i.e., with a resolution of

1 sq km) |

| | km with robust performance | | |
|---|---|---|---|
| Refer ences | http://cerea.enpc.fr /polyphemus/doc/P olyphemus-1.11-Guide.pdf

https://link.springer .com/article/10.100 7/s11869-019-00733-5/tables/1 | https://collaboration.cmc.ec.gc.c a/science/rpn/SEM/dossiers/200 9/seminaires/2009-05-08/Seminar_2009-05-08_Donald-Talbot.pdf

https://link.springer.com/article/ 10.1007/s11869-019-00733-5/tables/1

https://doi.org/10.5194/gmd-11-2609-2018

https://slideplayer.com/slide/65 40524/ | https://github.com/USEPA/CMAQ/tree/main/DOCS/Use rs_Guide

https://doi.org/10.5281/zenodo.5213949 |

1. Solazzo, Efisio, et al. "Ensemble modelling of surface-level ozone in Europe and North America for AQMEII." *Air Pollution Modeling and its Application XXII*. Springer, Dordrecht, 2014. 351-356.
2. Silveira, Carlos, Joana Ferreira, and Ana Isabel Miranda. "The challenges of air quality modelling when crossing multiple spatial scales." *Air Quality, Atmosphere & Health* 12.9 (2019): 1003-1017.

Sections 2.4 and Section 3.4: This appears to be a brute force, source-oriented source apportionment approach focusing on industrial emissions. However, it is not clear in these sections why this is being done, how does this "enrich the model validation findings"? For example, Lines 229-230 say that the model captures the spatial variability of the pollutants emitted from the industrial sector, but this was not evaluated directly.

This run section was run to simply test the model performance under varying emissions, and was not intended as a quantitative sensitivity analysis; sensitivity analysis was not the main focus of this study (that being model validation), it was included as we believed some preliminary analyses of this kind would strengthen the paper.

We have added two more scenarios on top of the "no industry" scenario (turning off emissions from the paper/pulp industry, and turning off emissions from the smelter/refinery industry) to strengthen this part of the paper.

In addition, we have performed additional analyses validating our modeled PM2.5 and NO2 against assimilated monthly ground-level PM2.5 and NO2 dataset products from ACAG (Atmospheric Composition Analysis Group) and CANUE (Canadian Urban Environmental Health Research Consortium) respectively; these datasets are widely used for air quality assessment and exposure studies. The results show fair agreements, indicating that the Polair3D model can be used for health impact analyses. Lastly, please note that we also reran the model for the month of January as we found some aberrations in some of the species in the ECCC offroad emissions (with the emissions themselves, not from our simulations). We reran the month using offroad emissions from April instead (everything else was kept the same), after checking that the offroad values did not vary significantly from month to month (in fact July and April for example were identical).

The meteorological modeling will impact the CTM results because that is driving the transport processes. The WRF configuration and evaluation information needs to be provided. Also, was two-way nesting done on the WRF simulations? If so, this means that the results from the 1km domains impact the results from the 3km domain and therefore comparing the model evaluation statistics for these two domains (i.e., Table 1) should be done with caution.

We have updated the manuscript and added more information regarding WRF configurations.

The WRF model was run separately for each domain. The model version 3.9.1.1 (Skamarock et al., 2008) was utilized. The modeling domain comprises four domains that have 27 km, 9 km, 3 km, and 1 km grid spacing, respectively, with two-way nesting. The number of vertical levels is 42 spanning from the surface to 100 hPa. Initial and lateral boundary conditions of meteorology were provided by the North American Regional Reanalysis (NARR; Mesinger et al., 2006) which is available at a 32 km grid spacing with 30 vertical levels. Each 30-hour forecast was initialized every 00 UTC and had a 6-hour spin-up time. Thus, the first 6 hours of forecasts are discarded and replaced with forecasts initiated with the previous cycle for overlapping times. Grid nudging was applied for horizontal wind, temperature, and humidity for vertical levels above the planetary boundary layer (PBL) height in the largest domain. Parameterization schemes used in the simulation are as follows: Purdue Lin scheme (Chen and Sun, 2002) for microphysics, the Rapid Radiative Transfer Model for GCMs (RRTMG) Shortwave and Longwave Schemes (Iacono et al., 2008), Mellor–Yamada–Janjic PBL scheme (Janjin and Zavisa, 1994), Grell–Devenyi ensemble scheme (Grell and Devenyi, 2002) for cumulus parameterization which was applied only to domain 1 and 2, Unified Noah land surface model (Chen and Dudhia, 2001), and a 3-category urban canopy model (Chen et al., 2011) for urban areas. This was added into the manuscript as well.

As meteorological data largely affect model results, we did indeed examine the meteorological input files from the Weather Research and Forecasting (WRF) model, and validated them

against observational temperature and wind data. This validation was done at all spatial resolutions, at hourly temporal resolution. Correlations, root mean square error (RMSE) and biases were calculated for temperature, wind speed, and wind directions. This was done at all meteorological stations in our modeling domain: those operated by Environment and Climate Change Canada (ECCC) and provinces.

As examples, Figures 1, and 2 present a comparison of WRF outputs and observational data at Montreal Trudeau airport at 1km and 3km resolutions. WRF output for October 1 to 14 2018 were compared against observations. Data were collected by ECCC and were downloaded from their publicly available website.

[Figure]

**Figure** 1. Time series plots of model (**at 1km resolution**) and in-situ observations of temperature (top left), zonal wind (bottom left), and meridional wind (bottom right) seen at **Montreal P.E. Trudeau airport**. Temperature correlation plot is also shown (top right). In all the time series plots, observation is indicated by blue, model by red, and the deltas by dashed grey.

[Figure]

**Figure** 2. Time series plots of model (**at 3km resolution**) and in-situ observations of temperature (top left), zonal wind (bottom left), and meridional wind (bottom right) seen at **Montreal P.E. Trudeau airport**. Temperature correlation plot is also shown (top right). In all the time series plots, observation is indicated by blue, model by red, and the deltas by dashed grey.

More discussion on the uncertainties related to emissions, meteorology, or chemistry would strengthen the paper. The authors provide a comparison of their evaluation results to three other studies in Section 3.3, but details about the potential sources of uncertainty in the model results are not provided, including whether or not those are similar to these other studies. Some examples include: Are there seasonal biases associated with chemical mechanisms or meteorology? Are the emissions inventories known to be biased high or low for a specific pollutant from a specific sector? The CTM results are the final results of an entire modeling framework, where uncertainties can be introduced at each stage.

**Additional discussion of uncertainties have been added to the manuscript (Section 3.1).** There have not been any generalized consensus regarding biases for Polair3D performance by season, although there have been reports of the model struggling during summer, especially for photochemically reactive species like ozone. Emissions of CH4 from oil and gas are reported to be underestimated in Canada (e.g., Lavoie et al., 2022; Conrad et al., 2023), and emissions of Criteria Air Contaminants (CACs) (which include nitrogen oxides, SO2, and VOCs) are also thought to be underestimated (Krzyzanowski, 2009). A sentence with this reference was also added to the discussion.

In addition, we have performed additional analyses validating our modeled PM2.5 and NO2 against assimilated monthly ground-level PM2.5 and NO2 dataset products from ACAG (Atmospheric Composition Analysis Group) and CANUE (Canadian Urban Environmental Health Research Consortium) respectively; these datasets are widely used for air quality assessment and exposure studies.

Minor Comments:

Line 7: add "horizontal" before "resolutions" here to indicate that these are the horizontal grid resolutions of the model.

This was fixed.

Line 8: The word "accurate" here implies that you are able to know to some level of certainty that the model is correct, does the data and evaluation you have available to you give you this level of certainty?

We have done a validation against the NAPS observation examining the model's spatial and seasonal variabilities of key pollutants. We have dropped the word "accurately" as we left out quantitative results (e.g., correlation and biases) in the abstract.

Line 37-40: It is not clear in here why Polair3D was selected, is there a specific motivation for using another CTM?

We have included a short discussion explaining our motivation.

Line 70: What is the vertical resolution or number of vertical grids in the model? Vertical resolution has a large impact on the simulation results.

We added the vertical resolution information.

Lines 75-77: Which chemical mechanism was used in SMOKE? Later, on line 103, the aerosol model is mentioned (AE6) but it would also be good to specify which chemical mechanism was used here. Did it require species mapping to run the SMOKE outputs with the Polair3D CTM?

CB05 and AE6 were the mechanisms used. Species mapping was required to run the smoke outputs with Polair3D.

For this study, we used the Carbon Bond mechanisms, CB05, as the gas-phase mechanism and AE6 aerosol scheme. The Polair3D model contains a Size-Composition Resolved Aerosol Model (SCRAM). Thus, the PM AE6 speciated SMOKE output must be incorporated into an input for SCRAM. This conversion is shown in Table 1.

The size distribution of the PM species was applied based on the SNAP (Selected Nomenclature for Air Pollution) sectors. For this, we compare the North American Industry Classification

System (NAICS) codes in the ECCC inventories with the SNAP using the Emission Inventory Guidebook of the European Environment Agency. The SNAP sectors include combustion in energy and transformation industries, non-industrial combustion plant, combustion in the manufacturing industry, production processes, extraction and distribution of fossil fuels and geothermal energy, solvent and other products, road transport other mobile sources and machinery, waste treatment and disposal, and agriculture. We applied a 5 bin size distribution to each species that was derived from the 10 bin SNAP size distribution values obtained from the Polair3D modelling team. For the Polair3D model, primary organic aerosol (POA) species also need to be divided into three categories based on pressure. Hence, first, we divided the POA species into low (POAlP), medium (POAmP), and high (POAhP) based on the percentages given by the Polyphemus team and then applied the 5 bin size distribution for each subspecies.

Table 1: SMOKE PM output conversion to Polyphemus PM input

| SMOKE PM Output Species | Polyphemus Input Species |
| --- | --- |
| PEC (elemental carbon) | PBC (0-4) (Black carbon) |
| PAL (aluminum) | |
| PCA (Calcium) | |
| PTI (Titanium) | |
| PFE (Iron) | PMD (0-3) (Mineral dust) |
| PSI (Silicon) | |
| PK (Potassium) | |
| PMG (Magnesium) | |
| PMN (Manganese) | |
| PMOTHR (PM not in other AE6 species) | |
| PMC (** PMC = PM10-PM2.5 ) | PMD_4 |
| PNH4 (Ammonium) | PNH4 (0-4) |
| PNO3 (Nitrate) | PNO3 (0-4) |
| PSO4 (Sulfate) | PSO4 (0-4) |
| POC (Organic Carbon) | PPOA (0-4) (Primary Organic Aerosol) -PPOAlP = 25% x PPOA -PPOAmP = 32% x PPOA -PPOAhP = 43% x PPOA |
| PNCOM (non-carbon organic matter) | |
| PCL (Chloride) | PHCL (0-4) |
| PNA (Sodium) | PNA (0-4) |

Line 78: U.S. EPA National Emissions Inventory (NEI) is the name of the inventory and, therefore, the N, E, and I should be capitalized (this occurs in other places in the paper as well).

This has been fixed.

Line 89-93: Was the U.S. EPA NEI platform used (e.g., spatial surrogates, species profiles, SMOKE scripts) or only the emissions input files? Also, specify the platform version number in addition to the year. The Technical Support Document (TSD) for the NEI could also be referenced here for how the different emissions sectors are modeled.

We did not use the entire NEI platform, only the spatial surrogate input files to generate spatial surrogates for our domain and inventory and species profiles in SMOKE.

Lines 94-102: There is a lot of detail in this section about the point sources but what about the mobile emissions, were those modeled using MOVES?

We used the SMOKE formatted mobile emissions inventory files from both Canadian and US inventories, and those inventories were prepared using emission factors from MOVES. MOVES was not used within SMOKE.

Line 122: Were only the metrics for evaluation used from Emery et al. (2017), or were the benchmark criteria for these metrics also used? If not, how are you defining success for the model?

The evaluation metric/methodology from Emery et al. (2017) was used. The "success" was discussed in the context of other studies.

Line 128: The sentence ending with "kept" feels like it is missing something after that word.

Changed to "kept the same."

Line 134: It is interesting to block out the lowest and highest concentrations in the spatial maps (white) because it seems like those concentrations might be the ones of interest, especially the high concentrations with respect to human health. Maybe plotting them on a log scale would help retain all of the pollutant concentrations but allow for visualization of the spatial gradients?

We did try log scale plots as well, but that also obscured details that we believe are important.